# Gradient Boosting Performs Gaussian Process Inference

**Aleksei Ustimenko**[*]
ShareChat
`research@aleksei.uk`

**Artem Beliakov**
Yandex Research, HSE University
`belyakov.arteom2015@gmail.com`

**Liudmila Prokhorenkova**
Yandex Research
`ostroumova-la@yandex.com`

## Abstract

This paper shows that gradient boosting based on symmetric decision trees can be equivalently reformulated as a kernel method that converges to the solution of a certain Kernel Ridge Regression problem. Thus, we obtain the convergence to a Gaussian Process' posterior mean, which, in turn, allows us to easily transform gradient boosting into a sampler from the posterior to provide better knowledge uncertainty estimates through Monte-Carlo estimation of the posterior variance. We show that the proposed sampler allows for better knowledge uncertainty estimates leading to improved out-of-domain detection.

## 1 Introduction

Gradient boosting (Friedman, 2001) is a classic machine learning algorithm successfully used for web search, recommendation systems, weather forecasting, and other problems (Roe et al., 2005; Caruana & Niculescu-Mizil, 2006; Richardson et al., 2007; Wu et al., 2010; Burges, 2010; Zhang & Haghani, 2015). In a nutshell, gradient boosting methods iteratively combine simple models (usually decision trees), minimizing a given loss function. Despite the recent success of neural approaches in various areas, gradient-boosted decision trees (GBDT) are still state-of-the-art algorithms for *tabular* datasets containing heterogeneous features (Gorishniy et al., 2021; Katzir et al., 2021).

This paper aims at a better theoretical understanding of GBDT methods for regression problems assuming the widely used RMSE loss function. First, we show that the gradient boosting with regularization can be reformulated as an optimization problem in some Reproducing Kernel Hilbert Space (RKHS) with implicitly defined kernel structure. After obtaining that connection between GBDT and kernel methods, we introduce a technique for sampling from prior Gaussian process distribution with the same kernel that defines RKHS so that the final output would converge to a sample from the Gaussian process posterior. Without this technique, we can view the output of GBDT as the mean function of the Gaussian process.

Importantly, our theoretical analysis assumes the regularized gradient boosting procedure (Algorithm 2) without any simplifications — we only need decision trees to be symmetric (oblivious) and properly randomized (Algorithm 1). These assumptions are non-restrictive and are satisfied in some popular gradient boosting implementations, e.g., CatBoost (Prokhorenkova et al., 2018).

Our experiments confirm that the proposed sampler from the Gaussian process posterior outperforms the previous approaches (Malinin et al., 2021) and gives better knowledge uncertainty estimates and improved out-of-domain detection.

---

[*]A major part of the work was done while working at Yandex Research.

## 2 BACKGROUND

Assume that we are given a distribution $\varrho$ over $X \times Y$, where $X \subset \mathbb{R}^d$ is called a feature space and $Y \subset \mathbb{R}$ — a target space. Further assume that we are given a dataset $\mathbf{z} = \{(x_i, y_i)\}_{i=1}^N \subset X \times Y$ of size $N \geq 1$ sampled i.i.d. from $\varrho$. Let us denote by $\rho(\mathrm{d}x) = \int_Y \mathrm{d}\varrho(\mathrm{d}x, \mathrm{d}y)$. W.l.o.g., we also assume that $X = \operatorname{supp} \rho = \overline{\{x \in \mathbb{R}^d : \forall \epsilon > 0, \rho(\{x' \in \mathbb{R}^d : \|x - x'\|_{\mathbb{R}^d} < \epsilon\}) > 0\}}$ which is a closed subset of $\mathbb{R}^d$. Moreover, for technical reasons, we assume that $\frac{1}{2N} \sum_{i=1}^N y_i^2 \leq R^2$ for some constant $R > 0$ almost surely, which can always be enforced by clipping. Throughout the paper, we also denote by $\mathbf{x}_N$ and $\mathbf{y}_N$ the matrix of all feature vectors and the vector of targets.

### 2.1 GRADIENT BOOSTED DECISION TREES

Given a loss function $L : \mathbb{R}^2 \to \mathbb{R}$, a classic gradient boosting algorithm (Friedman, 2001) iteratively combines weak learners (usually decision trees) to reduce the average loss over the train set $\mathbf{z}$: $\mathcal{L}(f) = \mathbb{E}_{\mathbf{z}}[L(f(x), y)]$. At each iteration $\tau$, the model is updated as: $f_\tau(x) = f_{\tau-1}(x) + \epsilon w_\tau(x)$, where $w_\tau(\cdot) \in \mathcal{W}$ is a weak learner chosen from some family of functions $\mathcal{W}$, and $\epsilon$ is a learning rate. The weak learner $w_\tau$ is usually chosen to approximate the negative gradient of the loss function $-g_\tau(x, y) := -\frac{\partial L(s,y)}{\partial s}\big|_{s=f_{\tau-1}(x)}$:

$$w_\tau = \underset{w \in \mathcal{W}}{\arg\min} \, \mathbb{E}_{\mathbf{z}}\big[\big(-g_\tau(x, y) - w(x)\big)^2\big]. \tag{1}$$

The family $\mathcal{W}$ usually consists of decision trees. In this case, the algorithm is called GBDT (Gradient Boosted Decision Trees). A decision tree is a model that recursively partitions the feature space into disjoint regions called leaves. Each leaf $R_j$ of the tree is assigned to a value, which is the estimated response $y$ in the corresponding region. Thus, we can write $w(x) = \sum_{j=1}^d \theta_j \mathbf{1}_{\{x \in R_j\}}$, so the decision tree is a linear function of the leaf values $\theta_j$.

A recent paper Ustimenko & Prokhorenkova (2021) proposes a modification of classic stochastic gradient boosting (SGB) called *Stochastic Gradient Langevin Boosting* (SGLB). SGLB combines gradient boosting with stochastic gradient Langevin dynamics to achieve global convergence even for non-convex loss functions. As a result, the obtained algorithm provably converges to some stationary distribution (invariant measure) concentrated near the global optimum of the loss function. We mention this method because it samples from similar distribution as our method but with a different kernel.

### 2.2 ESTIMATING UNCERTAINTY

In addition to the predictive quality, it is often important to detect when the system is uncertain and can be mistaken. For this, different measures of *uncertainty* can be used. There are two main sources of uncertainty: *data uncertainty* (a.k.a. aleatoric uncertainty) and *knowledge uncertainty* (a.k.a. epistemic uncertainty). Data uncertainty arises due to the inherent complexity of the data, such as additive noise or overlapping classes. For instance, if the target is distributed as $y|x \sim \mathcal{N}(f(x), \sigma^2(x))$, then $\sigma(x)$ reflects the level of data uncertainty. This uncertainty can be assessed if the model is probabilistic.

Knowledge uncertainty arises when the model gets input from a region either sparsely covered by or far from the training data. Since the model does not have enough data in this region, it will likely make a mistake. A standard approach to estimating knowledge uncertainty is based on *ensembles* (Gal, 2016; Malinin, 2019). Assume that we have trained an ensemble of several independent models. If all the models understand an input (low knowledge uncertainty), they will give similar predictions. However, for out-of-domain examples (high knowledge uncertainty), the models are likely to provide diverse predictions. For regression tasks, one can obtain knowledge uncertainty by measuring the variance of the predictions provided by multiple models (Malinin, 2019).

Such ensemble-based approaches are standard for neural networks (Lakshminarayanan et al., 2017). Recently, ensembles were also tested for GBDT models (Malinin et al., 2021). The authors consider two ways of generating ensembles: ensembles of independent SGB models and ensembles of independent SGLB models. While empirically methods are very similar, SGLB has better theoretical properties: the convergence of parameters to the stationary distribution allows one to sample models

from a particular posterior distribution. One drawback of SGLB is that its convergence rate is unknown, as the proof is asymptotic. However, the convergence rate can be upper bounded by that of Stochastic Gradient Langevin Dynamics for log-concave functions, e.g., Zou et al. (2021), which is not dimension-free. In contrast, our rate is dimension-free and scales linearly with inverse precision.

## 2.3 GAUSSIAN PROCESS INFERENCE

In this section, we briefly describe the basics of Bayesian inference with Gaussian processes that is closely related to our analysis. A random variable $f$ with values in $L_2(\rho)$ is said to be a Gaussian process $f \sim \mathcal{GP}(f_0, \sigma^2 \mathcal{K} + \delta^2 \mathrm{Id}_{L_2})$ with covariance defined via a kernel $\mathcal{K}(x, x') = \mathrm{cov}(f(x), f(x'))$ and mean value $f_0 \in L_2(\rho)$ iff $\forall g \in L_2(\rho)$ we have

$$\int_X f(x)g(x)\rho(\mathrm{d}x) \sim \mathcal{N}\Big(\int_X f_0(x)g(x)\rho(\mathrm{d}x), \sigma^2 \int_{X \times X} g(x)\mathcal{K}(x, x')g(x')\rho(\mathrm{d}x) \otimes \rho(\mathrm{d}x') + \delta^2 \|g\|_{L_2}^2\Big).$$

A typical Gaussian Process setup is to assume that the target $y|x$ is distributed as $\mathcal{GP}(\mathbb{0}_{L_2(\rho)}, \sigma^2 \mathcal{K} + \delta^2 \mathrm{Id}_{L_2})$ for some kernel function[1] $\mathcal{K}(x, x')$ with scales $\sigma > 0$ and $\delta > 0$:

$$y|x = f(x), \;\; f \sim \mathcal{GP}(\mathbb{0}_{L_2(\rho)}, \sigma^2 \mathcal{K} + \delta^2 \mathrm{Id}_{L_2}).$$

The posterior distribution $f(x)|x, \mathbf{x}_N, \mathbf{y}_N$ is again a Gaussian Process $\mathcal{GP}(f_*, \sigma^2 \widetilde{\mathcal{K}} + \delta^2 \mathrm{Id}_{L_2})$ with mean and covariance given by (see Rasmussen & Williams (2006)):

$$f_*^\lambda(x) = \mathcal{K}(x, \mathbf{x}_N)\big(\mathcal{K}(\mathbf{x}_N, \mathbf{x}_N) + \lambda I_N\big)^{-1}\mathbf{y}_N,$$
$$\widetilde{\mathcal{K}}(x, x) = \mathcal{K}(x, x) - \mathcal{K}(x, \mathbf{x}_N) \cdot \big(\mathcal{K}(\mathbf{x}_N, \mathbf{x}_N) + \lambda I_N\big)^{-1}\mathcal{K}(\mathbf{x}_N, x).$$

with $\lambda = \frac{\delta^2}{\sigma^2}$. To estimate the posterior mean $f_*^\lambda(x) = \int_{\mathbb{R}} f p(f|x, \mathbf{x}_N, \mathbf{y}_N) \, \mathrm{d}f$, we can use the maximum a posteriori probability estimate (MAP) — the only solution for the following Kernel Ridge Regression (KRR) problem:

$$L(f) = \frac{1}{2N} \sum_{i=1}^{N} \big(f(x_i) - y_i\big)^2 + \frac{\lambda}{2N}\|f\|_{\mathcal{H}}^2 \to \min_{f \in \mathcal{H}},$$

where $\mathcal{H} = \overline{\mathrm{span}\{\mathcal{K}(\cdot, x)|x \in X\}} \subset L_2(\rho)$ is the reproducing kernel Hilbert space (RKHS) for the kernel $\mathcal{K}(\cdot, \cdot)$ and $L(f) \to \min_{f \in \mathcal{H}}$ means that we are seeking minimizers of this function $L(f)$. We refer to Appendix D for the details on KRR and RKHS.

To solve the KRR problem, one can apply the gradient descent (GD) method in the functional space:

$$f_{\tau+1} = (1 - \frac{\lambda \epsilon}{N})f_\tau - \epsilon \frac{1}{N} \sum_{i=1}^{N} (f_\tau(x_i) - y_i)\mathcal{K}_{x_i}, \; f_0 = \mathbb{0}_{L_2(\rho)}, \qquad (2)$$

where $\epsilon > 0$ is a learning rate.

Since this objective is strongly convex due to the regularization, the gradient descent rapidly converges to the unique optimum:

$$f_*^\lambda = \lim_{\tau \to \infty} f_\tau = \mathcal{K}(\cdot, \mathbf{x}_N)\big(\mathcal{K}(\mathbf{x}_N, \mathbf{x}_N) + \lambda I_N\big)^{-1}\mathbf{y}_N,$$

see Appendices C and E for the details.

Gradient descent guides $f_\tau$ to the posterior mean $f_*^\lambda$ of the Gaussian Process with kernel $\sigma^2 \mathcal{K} + \delta^2 \mathrm{Id}_{L_2}$. To obtain the posterior variance estimate $\widetilde{\mathcal{K}}(x, x)$ for any $x$, one can use sampling and introduce a source of randomness in the above iterative scheme as follows:

1. sample $f^{init} \sim \mathcal{GP}(\mathbb{0}_{L_2(\rho)}, \sigma^2 \mathcal{K} + \delta^2 \mathrm{Id}_{L_2})$;
2. set new labels $\mathbf{y}_N^{new} = \mathbf{y}_N - f^{init}(\mathbf{x}_N)$;
3. fit GD $f_\tau(\cdot)$ on $\mathbf{y}_N^{new}$ assuming $F_0(\cdot) = \mathbb{0}_{L_2(\rho)}$;
4. output $f^{init}(\cdot) + f_\tau(\cdot)$ as final model.

---

[1] $\mathcal{K}(x, x')$ is a kernel function if $[\mathcal{K}(x_i, x_j)]_{i=1, j=1}^{N,N} \geq 0$ for any $N \geq 1$ and any $x_i \in X$ almost surely.

This method is known as Sample-then-Optimize (Matthews et al., 2017) and is widely adopted for Bayesian inference. As $\tau \to \infty$, we get $f^{init} + f_\infty$ distributed as a Gaussian Process posterior with the desired mean and variance. Formally, the following result holds:

**Lemma 2.1.** $f^{init} + f_\infty$ *follows the Gaussian Process posterior* $\mathcal{GP}(f_*^\lambda, \sigma^2 \widetilde{\mathcal{K}} + \delta^2 \mathrm{Id}_{L_2})$ *with:*

$$f_*^\lambda(x) = \mathcal{K}(x, \mathbf{x}_N) \big( \mathcal{K}(\mathbf{x}_N, \mathbf{x}_N) + \lambda I_N \big)^{-1} \mathbf{y}_N \,,$$

$$\widetilde{\mathcal{K}}(x, x) = \mathcal{K}(x, x) - \mathcal{K}(x, \mathbf{x}_N) \big( \mathcal{K}(\mathbf{x}_N, \mathbf{x}_N) + \lambda I_N \big)^{-1} \mathcal{K}(\mathbf{x}_N, x) \,.$$

# 3 EVOLUTION OF GBDT IN RKHS

## 3.1 PRELIMINARIES

In our analysis, we assume that we are given a finite set $\mathcal{V}$ of weak learners used for the gradient boosting.[2] Each $\nu$ corresponds to a decision tree that defines a partition of the feature space into disjoint regions (leaves). For each $\nu \in \mathcal{V}$, we denote the number of leaves in the tree by $L_\nu \geq 1$. Also, let $\phi_\nu : X \to \{0, 1\}^{L_\nu}$ be a mapping that maps $x$ to the vector indicating its leaf index in the tree $\nu$. This mapping defines a decomposition of $X$ into the disjoint union: $X = \cup_{j=1}^{L_\nu} \big\{ x \in X \big| \phi_\nu^{(j)}(x) = 1 \big\}$. Having $\phi_\nu$, we define a weak learner associated with it as $x \mapsto \langle \theta, \phi_\nu(x) \rangle_{\mathbb{R}^{L_\nu}}$ for any choice of $\theta \in \mathbb{R}^{L_\nu}$ which we refer to as 'leaf values'. In other words, $\theta$ corresponds to predictions assigned to each region of the space defined by $\nu$.

Let us define a linear space $\mathcal{F} \subset L_2(\rho)$ of all possible ensembles of trees from $\mathcal{V}$:

$$\mathcal{F} = \mathrm{span} \big\{ \phi_\nu^{(j)}(\cdot) : X \to \{0, 1\} \big| \nu \in \mathcal{V}, j \in \{1, \dots, L_\nu\} \big\} \,.$$

We note that the space $\mathcal{F}$ can be data-dependent since $\mathcal{V}$ may depend on $\mathbf{z}$, but we omit this dependence in the notation for simplicity. Note that we do not take the closure w.r.t. the topology of $L_2(\rho)$ since we assume that $\mathcal{V}$ is finite and therefore $\mathcal{F}$ is finite-dimensional and thus topologically closed.

## 3.2 GBDT ALGORITHM UNDER CONSIDERATION

Our theoretical analysis holds for classic GBDT algorithms discussed in Section 2.1 equipped with regularization from Ustimenko & Prokhorenkova (2021). The only requirement we need is that the procedure of choosing each new tree has to be properly randomized. Let us discuss a tree selection algorithm that we assume in our analysis.

Each new tree approximates the gradients of the loss function with respect to the current predictions of the model. Since we consider the RMSE loss function, the gradients are proportional to the residuals $r_j = y_j - f(x_j)$, where $f$ is the currently built model. The tree structure is defined by the features and the corresponding thresholds used to split the space. The analysis in this paper assumes the *SampleTree* procedure (see Algorithm 1), which is a classic approach equipped with proper randomization. *SampleTree* builds an *oblivious* decision tree (Prokhorenkova et al., 2018), i.e., all nodes at a given level share the same splitting criterion (feature and threshold).[3] To limit the number of candidate splits, each feature is quantized into $n + 1$ bins. In other words, for each feature, we have $n$ thresholds that can be chosen arbitrarily.[4] The maximum tree depth is limited by $m$. Recall that we denote the set of all possible tree structures by $\mathcal{V}$.

We build the tree in a top-down greedy manner. At each step, we choose one split among all the remaining candidates based on the following *score* defined for $\nu \in \mathcal{V}$ and residuals $r$:

$$D(\nu, r) := \frac{1}{N} \sum_{j=1}^{L_v} \frac{\left( \sum_{i=1}^{N} \phi_\nu^{(j)}(x_i) \, r_i \right)^2}{\sum_{i=1}^{N} \phi_\nu^{(j)}(x_i)} \,. \tag{3}$$

---

[2]The finiteness of $\mathcal{V}$ is important for our analysis, and it usually holds in practice, see Section 3.2.

[3]In fact, the procedure can be extended to arbitrary trees, but this would over-complicate formulation of the algorithm and would not change the space of tree ensembles as any non-symmetric tree can be represented as a sum of symmetric ones.

[4]A standard approach is to quantize the feature such that all $n + 1$ buckets have approximately the same number of training samples.

**Algorithm 1** SampleTree($r; m, n, \beta$)

**input:** residuals $r = (r_i)_{i=1}^N$
**output:** oblivious tree structure $\nu \in \mathcal{V}$
**hyper-parameters:** number of feature splits $n$, max. tree depth $m$, random strength $\beta \in [0, \infty)$
**definitions:**
$\mathcal{S} = \big\{(j,k)\big| j \in \{1, \ldots, d\}, k \in \{1, \ldots, n\}\big\}$ — indices of all possible splits

**instructions:**
initialize $i = 0$, $\nu_0 = \emptyset$, $\mathcal{S}^{(0)} = \mathcal{S}$
**repeat**
$\quad$ sample $(u_i(s))_{s \in \mathcal{S}^{(i)}} \sim \mathrm{U}\big([0,1]^{nd-i}\big)$
$\quad$ choose next split as $\{s_{i+1}\} = \underset{s \in \mathcal{S}^{(i)}}{\arg\max}\Big(D\big((\nu_i, s), r\big) - \beta \log(-\log u_i(s))\Big)$
$\quad$ update tree: $\nu_{i+1} = (\nu_i, s_{i+1})$
$\quad$ update candidate splits: $\mathcal{S}^{(i+1)} = \mathcal{S}^{(i)} \setminus \{s_{i+1}\}$

$\quad$ $i = i + 1$
**until** $i \geq m$ **or** $\mathcal{S}^{(i)} = \emptyset$
**return:** $\nu_i$

**Algorithm 2** TrainGBDT($\mathbf{z}; \epsilon, T, m, n, \beta, \lambda$)

**input:** dataset $\mathbf{z} = (\mathbf{x}_N, \mathbf{y}_N)$
**hyper-parameters:** learning rate $\epsilon > 0$, regularization $\lambda > 0$, iterations of boosting $T$, parameters of SampleTree $m, n, \beta$

**instructions:**
initialize $\tau = 0$, $f_0(\cdot) = \mathbb{0}_{L_2(\rho)}$
**repeat**
$\quad$ $r_\tau = \mathbf{y}_N - f_\tau(\mathbf{x}_N)$ — compute residuals
$\quad$ $\nu_\tau = \mathrm{SampleTree}(r_\tau; m, n, \beta)$ — construct a tree
$\quad$ $\theta_\tau = \Big(\frac{\sum_{i=1}^N \phi_{\nu_\tau}^{(j)}(x_i) r_\tau^{(i)}}{\sum_{i=1}^N \phi_{\nu_\tau}^{(j)}(x_i)}\Big)_{j=1}^{L_{\nu_\tau}}$ — set values in leaves
$\quad$ $f_{\tau+1}(\cdot) = (1 - \frac{\lambda\epsilon}{N}) f_\tau(\cdot) + \epsilon\big\langle \phi_{\nu_\tau}(\cdot), \theta_\tau \big\rangle_{\mathbb{R}^{L_{\nu_\tau}}}$ — update model
$\quad$ $\tau = \tau + 1$
**until** $\tau \geq T$
**return:** $f_T(\cdot)$

In classic gradient boosting, one builds a tree recursively by choosing such split $s$ that maximizes the score $D((\nu_i, s), r)$ (Ibragimov & Gusev, 2019).[5] Random noise is often added to the scores to improve generalization. In *SampleTree*, we choose a split that maximizes

$$D\big((\nu_i, s), r\big) + \varepsilon, \text{ where } \varepsilon \sim \mathrm{Gumbel}(0, \beta). \tag{4}$$

Here $\beta$ is random strength: $\beta = 0$ gives the standard greedy approach, while $\beta \to \infty$ gives the uniform distribution among all possible split candidates.

To sum up, *SampleTree* is a classic oblivious tree construction but with added random noise. We do this to make the distribution of trees regular in a certain sense: roughly speaking, the distributions should stabilize with iterations by converging to some fixed distribution.

Given the algorithm *SampleTree*, we describe the gradient boosting procedure assumed in our analysis in Algorithm 2. It is a classic GBDT algorithm but with the update rule $f_{\tau+1}(\cdot) = (1 - \lambda\epsilon/N) f_\tau(\cdot) + \epsilon w_\tau(x)$. In other words, we shrink the model at each iteration, which serves as regularization (Ustimenko & Prokhorenkova, 2021). Such shrinkage is available, e.g., in the CatBoost library.

### 3.3 Distribution of trees

The *SampleTree* algorithm induces a local family of distributions $p(\cdot | f, \beta)$ for each $f \in \mathcal{F}$:

$$p(\mathrm{d}\nu | f, \beta) = \mathbb{P}\Big(\big\{\mathrm{SampleTree}\big(\mathbf{y}_N - f(\mathbf{x}_N); m, n, \beta\big) \in \mathrm{d}\nu\big\}\Big).$$

*Remark* 3.1. Lemma D.5 ensure that such distribution coincides with the one where we use $f_*(\mathbf{x}_N)$ instead of $\mathbf{y}_N$. This is due to the fact that $D\big(\nu, \mathbf{y}_N - f(\mathbf{x}_N)\big) = D\big(\nu, f_* - f(\mathbf{x}_N)\big) \forall \nu \in \mathcal{V}, f \in \mathcal{F}$.

The following lemma describes the distribution $p(\mathrm{d}\nu | f, \beta)$, see Appendix F for the proof.

**Lemma 3.2.** *(Probability of a tree)*[6]

$$p(\nu | f, \beta) = \sum_{\varsigma \in \mathcal{P}_m} \prod_{i=1}^m \frac{e^{\frac{D(\nu_{\varsigma, i}, r)}{\beta}}}{\sum_{s \in \mathcal{S} \setminus \nu_{\varsigma, i-1}} e^{\frac{D((\nu_{\varsigma, i-1}, s), r)}{\beta}}},$$

---

[5] Maximizing (3) is equivalent to minimizing the squared error between the residuals and the mean values in the leaves.

[6] Note that for oblivious decision trees, changing the order of splits does not affect the obtained partition. Hence, we assume that each tree is defined by an unordered set of splits.

*where the sum is over all permutations* $\varsigma \in \mathcal{P}_m$, $\nu_{\varsigma,i} = (s_{\varsigma(1)}, \ldots, s_{\varsigma(i)})$, *and* $\nu = (s_1, \ldots, s_m)$.

Let us define the stationary distribution of trees as $\pi(\cdot) = \lim_{\beta \to \infty} p(\cdot | f, \beta)$. It follows from Remark 3.1 that we also have $\pi(\cdot) = p(\cdot | f_*, \beta)$.

**Corollary 3.3.** *(Stationary distribution is the uniform distribution over tree structures)*
*We have* $\pi(\mathrm{d}\nu) = |\mathrm{d}\nu| / \binom{nd}{m}$, *where* $\binom{nd}{m} = \frac{(nd)!}{(nd-m)!m!}$.

## 3.4 RKHS STRUCTURE

In this section, we describe the evolution of GBDT in a certain Reproducing Kernel Hilbert Space (RKHS). Even though the problem is finite dimensional, treating it as functional regression is more beneficial as dimension of the ensembles space grows rapidly and therefore we want to obtain dimension-free constants which is impossible if we treat it as finite dimensional optimization problem. Let us start with defining necessary kernels. For convenience, we also provide a diagram illustrating the introduced kernels and relations between them in Appendix A.

**Definition 3.4.** A weak learner's kernel $k_\nu(\cdot, \cdot)$ is a kernel function associated with a tree structure $\nu \in \mathcal{V}$ which can be defined as:

$$k_\nu(x, x') = \sum_{j=1}^{L_\nu} w_\nu^{(j)} \phi_\nu^{(j)}(x) \phi_\nu^{(j)}(x'), \text{ where } w_\nu^{(j)} = \frac{N}{\max\{N_\nu^{(j)}, 1\}}, N_\nu^{(j)} = \sum_{i=1}^{N} \phi_\nu^{(j)}(x_i).$$

This weak learner's kernel is a building block for any other possible kernel in boosting and is used to define the iterations of the boosting algorithm analytically.

**Definition 3.5.** We also define a *greedy* kernel of the gradient boosting algorithm as follows:

$$\mathcal{K}_f(x, x') = \sum_{\nu \in \mathcal{V}} k_\nu(x, x') p(\nu | f, \beta).$$

This greedy kernel is a kernel that guides the GBDT iterations, i.e., we can think of each iteration as SGD with a kernel from 3.5, and 3.4 is used as a stochastic gradient estimator of the Fréchet derivative in RKHS defined by the kernel from 3.5.

**Definition 3.6.** Finally, there is a *stationary* kernel $\mathcal{K}(x, x')$ that is independent from $f$:

$$\mathcal{K}(x, x') = \sum_{\nu \in \mathcal{V}} k_\nu(x, x') \pi(\nu),$$

which we call a *prior* kernel of the gradient boosting.

This kernel defines the limiting solution since the gradient projection on RKHS converges to zero, and thus 3.5 converges to 3.6.

Note that $\mathcal{F} = \mathrm{span}\{\mathcal{K}(\cdot, x) \mid x \in \mathcal{X}\}$. Having the space of functions $\mathcal{F}$, we define RKHS structure $\mathcal{H} = (\mathcal{F}, \langle \cdot, \cdot \rangle_{\mathcal{H}})$ on it using a scalar product defined as

$$\langle f, \mathcal{K}(\cdot, x) \rangle_{\mathcal{H}} = f(x).$$

Now, let us define the empirical error of a model $f$:

$$L(f, \lambda) = \frac{1}{2N} \|y_N - f(\mathbf{x}_N)\|_{\mathbb{R}^N}^2 + \frac{\lambda}{2N} \|f\|_{\mathcal{H}}^2.$$

Then, we define $V(f, \lambda) = L(f, \lambda) - \inf_{f' \in \mathcal{F}} L(f', \lambda)$. Let us also define the following functions: $f_*^\lambda \in \arg\min_{f \in \mathcal{F}} V(f, \lambda)$ and

$$f_* = \lim_{\lambda \to 0} f_*^\lambda \in \arg\min_{f \in \mathcal{F}} V(f), \text{ where } V(f) = V(f, 0).$$

It is known that such $f_*$ exists and is unique since the set of all solutions is convex, and therefore there is a unique minimizer of the norm $\| \cdot \|_{\mathcal{H}}$. Finally, the following lemma gives the formula of the GBDT iteration in terms of kernels in Lemma 3.7 which will be useful in proving our results. See Appendix D for the proofs.

**Lemma 3.7.** *Iterations* $f_\tau$ *of Gradient Boosting (Algorithm 2) can be written in the form:*

$$f_{\tau+1} = (1 - \frac{\lambda\epsilon}{N}) f_\tau + \frac{\epsilon}{N} k_{\nu_\tau}(\cdot, \mathbf{x}_N) [f_*(\mathbf{x}_N) - f_\tau(\mathbf{x}_N)], \nu_\tau \sim p(\nu | f_\tau, \beta).$$

## 3.5 KERNEL GRADIENT BOOSTING CONVERGENCE TO KRR

Consider the sequence $\{f_\tau\}_{\tau \in \mathbb{N}}$ generated by the gradient boosting algorithm. Its evolution is described by Lemma 3.7. The following theorem estimates the expected (w.r.t. the randomness of tree selection) empirical error of $f_T$ relative to the best possible ensemble. The full statement of the theorem and its proof can be found in Appendix G.

**Theorem 3.8.** *Assume that $\beta, T_1$ are sufficiently large and $\epsilon$ is sufficiently small (see Appendix G). Then, $\forall T \geq T_1$,*

$$\mathbb{E} V(f_T, \lambda) \leq \mathcal{O}\Big( e^{-\mathcal{O}\left( \frac{\epsilon(T-T_1)}{N} \right)} + \frac{\lambda^2 \epsilon}{N^2} + \epsilon + \frac{\lambda}{\beta N^2} \Big).$$

**Corollary 3.9.** *(Convergence to the solution of the KRR problem) Under the assumptions of the previous theorem, we have the following dimension-free bound:*

$$\mathbb{E}\|f_T - f_*^\lambda\|_{L_2}^2 \leq \mathcal{O}\Big( e^{-\mathcal{O}\left( \frac{\epsilon(T-T_1)}{N} \right)} + \frac{\lambda^2 \epsilon}{N} + N\epsilon + \frac{\lambda}{\beta N} \Big).$$

This bound is dimension-free thanks to functional treatment and exponentially decaying to small value with iterations and therefore justifies the observed rapid convergence of gradient boosting algorithms in practice even though dimension of space $\mathcal{H}$ is enormous.

## 4 GAUSSIAN INFERENCE

So far, the main result of the paper proved in Section 3.5 shows that Algorithm 2 solves the Kernel Ridge Regression problem, which can be interpreted as learning Gaussian Process posterior mean $f_*^\lambda$ under the assumption that $f \sim \mathcal{GP}(0, \sigma^2 \mathcal{K} + \delta^2 \mathrm{Id}_{L_2})$ where $\lambda = \frac{\sigma^2}{\delta^2}$. I.e., Algorithm 2 does not give us the posterior variance. Still, as mentioned earlier, we can estimate the posterior variance through Monte-Carlo sampling in a sample-then-optimize way. For that, we need to somehow sample from the prior distribution $\mathcal{GP}(0, \sigma^2 \mathcal{K} + \delta^2 \mathrm{Id}_{L_2})$.

### 4.1 PRIOR SAMPLING

We introduce Algorithm 3 for sampling from the prior distribution. *SamplePrior* generates an ensemble of random trees (with random splits and random values in leaves). Note that while being random, the tree structure depends on the dataset features $\mathbf{x}_N$ since candidate splits are based on $\mathbf{x}_N$.

We first note that the process $h_T(\cdot)$ is centered with covariance operator $\mathcal{K}$:

$$\mathbb{E} h_T(x) = 0 \ \ \forall x \in X \,,$$
$$\mathbb{E} h_T(x) h_T(y) = \mathcal{K}(x, y) \ \ \forall x, y \in X \,. \tag{5}$$

Then, we show that $h_T(\cdot)$ converges to the Gaussian Process in the limit.

**Lemma 4.1.** *The following convergence holds almost surely in $x \in X$:*

$$h_T(\cdot) \xrightarrow[T \to \infty]{} \mathcal{GP}\big( \mathbb{0}_{L_2(\rho)}, \mathcal{K} \big) \,.$$

---

**Algorithm 3** SamplePrior$(T, m, n)$

**hyper-parameters:** number of iterations $T$, parameters of SampleTree $m, n$

**instructions:**
initialize $\tau = 0$, $h_0(x) = 0$
**repeat**
  $\nu_\tau = \text{SampleTree}(\mathbb{0}_{\mathbb{R}^N}; m, n, 1)$ — sample random tree
  $\theta_\tau \sim \mathcal{N}\Big( \mathbb{0}_{\mathbb{R}^{L_{\nu_\tau}}}, \mathrm{diag}\Big( \frac{N}{\max\{N_{\nu_\tau}^{(j)}, 1\}} : j \in$
  $\{1, \ldots, L_{\nu_\tau}\} \Big) \Big)$ — generate random values
  in leaves
  $h_{\tau+1}(\cdot) = h_\tau(\cdot) + \frac{1}{\sqrt{T}} \langle \phi_{\nu_\tau}(\cdot), \theta_\tau \rangle_{\mathbb{R}^{L_{\nu_\tau}}}$ —
  update model
  $\tau = \tau + 1$
**until** $\tau \geq T$
**return:** $h_T(\cdot)$

---

### 4.2 POSTERIOR SAMPLING

Now we are ready to introduce Algorithm 4 for sampling from the posterior. The procedure is simple: we first perform $T_0$ iterations of SamplePrior to obtain a function $h_{T_0}(\cdot)$ and then we train a standard GBDT model $f_{T_1}(\cdot)$ approximating $\mathbf{y}_N - \sigma h_{T_0}(\mathbf{x}_N) + \mathcal{N}(\mathbb{0}_N, \delta^2 I_N)$. Our final model is $\sigma h_{T_0}(\cdot) + f_{T_1}(\cdot)$.

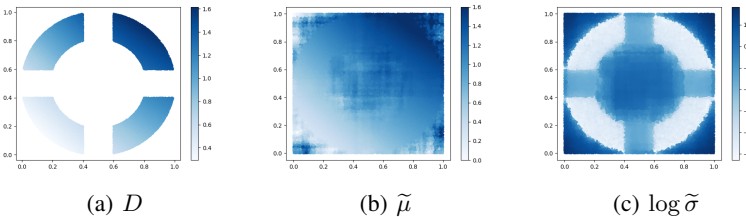

Figure 1: KGB on a synthetic dataset.

We further refer to this procedure as *SamplePosterior* or *KGB* (Kernel Gradient Boosting) for brevity. Denote

$$h_\infty = \lim_{T_0 \to \infty} h_{T_0} \,, f_\infty = \lim f_{T_1} \,,$$

where the first limit is with respect to the point-wise convergence of stochastic processes and the second one with respect to $L_2(\rho)$ convergence.

The following theorem shows that KGB indeed samples from the desired posterior. The proof directly follows from Lemmas 4.1 and 2.1.

---

**Algorithm 4** SamplePosterior($\mathbf{z}; \epsilon, T_1, T_0, m, n, \beta, \sigma, \delta$)

**input:** dataset $\mathbf{z} = (\mathbf{x}_N, \mathbf{y}_N)$
**hyper-parameters:** learning rate $\epsilon > 0$, boosting iteration $T_1$, SamplePrior iterations $T_0$, parameters of SampleTree $m, n, \beta$, kernel scale $\sigma > 0$ (default $\sigma = 1$), noise scale $\delta > 0$ (default: $\delta = 0.01$)
**instructions:**
$h_{T_0}(\cdot) = \text{SamplePrior}(T_0, m, n)$
$\mathbf{y}_N^{new} = \mathbf{y}_N - \sigma h_{T_0}(\mathbf{x}_N) + \mathcal{N}(\mathbb{0}_N, \delta^2 I_N)$
$f_{T_1}(\cdot) = \text{TrainGBDT}\big((\mathbf{x}_N, \mathbf{y}_N^{new}); \epsilon, T_1, m, n, \beta, \frac{\delta^2}{\sigma^2}\big)$

**return:** $\sigma h_{T_0}(\cdot) + f_{T_1}(\cdot)$

---

**Theorem 4.2.** *In the limit, the output of Algorithm 4 follows the Gaussian process posterior:*

$$\sigma h_\infty(\cdot) + f_\infty(\cdot) + \mathcal{N}(0, \delta^2) \sim \mathcal{GP}\big(\widetilde{f}, \widetilde{\mathcal{K}}\big)$$

*with mean* $\widetilde{f}(x) = \mathcal{K}(x, \mathbf{x}_N)\big(\mathcal{K}(\mathbf{x}_N, \mathbf{x}_N) + \lambda I_N\big)^{-1} \mathbf{y}_N$ *and covariance* $\widetilde{\mathcal{K}}(x, x) = \delta^2 + \sigma^2\big(\mathcal{K}(x, x) - \mathcal{K}(x, \mathbf{x}_N)\big(\mathcal{K}(\mathbf{x}_N, \mathbf{x}_N) + \lambda I_N\big)^{-1} \mathcal{K}(\mathbf{x}_N, x)\big)$.

## 5 EXPERIMENTS

This section empirically evaluates the proposed KGB algorithm and shows that it indeed allows for better knowledge uncertainty estimates.

**Synthetic experiment** To illustrate the KGB algorithm in a controllable setting, we first conduct a synthetic experiment. For this, we defined the feature distribution as uniform over $D = \{(x, y) \in [0, 1]^2 : \frac{1}{10} \le (x - \frac{1}{2})^2 - (y - \frac{1}{2})^2 \le \frac{1}{4} \land (x \le \frac{2}{5} \lor x \ge \frac{3}{5}) \land (y \le \frac{2}{5} \lor y \ge \frac{3}{5})\}$. We sample 10K points from $U([0, 1]^2)$ and take into the train set only those that fall into $D$. The target is defined as $f(x, y) = x + y$. Figure 1(a) illustrates the training dataset colored with the target values. For evaluation, we take the same 10K points without restricting them to $D$.

For KGB, we fix $\epsilon = 0.3$, $T_0 = 100$, $T_1 = 900$, $\sigma = 10^{-2}$, $\delta = 10^{-4}$ $\beta = 0.1$, $m = 4$, $n = 64$, and sampled 100 KGB models. Figure 1(b) shows the estimated by Monte-Carlo posterior mean $\widetilde{\mu}$. On Figure 1(c), we show $\log \widetilde{\sigma}$, where $\widetilde{\sigma}^2$ is the posterior variance estimated by Monte-Carlo. One can see that the posterior variance is small in-domain and grows when we move outside the dataset $D$, as desired.

**Experiment on real datasets** Uncertainty estimates for GBDTs have been previously analyzed by Malinin et al. (2021). Our experiments on real datasets closely follow their setup, and we compare the proposed KGB with SGB, SGLB, and their ensembles. For the experiments, we use several standard regression datasets (Gal & Ghahramani, 2016). The implementation details can be found in Appendix H. The code of our experiments can be found on GitHub.[7]

---

[7]https://github.com/TakeOver/Gradient-Boosting-Performs-Gaussian-Process-Inference

Table 1: Predictive performance, RMSE

| Dataset | Single | | | Ensemble | | |
|---------|--------|------|------|----------|------|------|
| | SGB | SGLB | KGB | SGB | SGLB | KGB |
| Boston | 3.06 | 3.12 | **2.81** | 3.04 | 3.10 | **2.82** |
| Concrete | 5.21 | 5.11 | **4.36** | 5.21 | 5.10 | **4.30** |
| Energy | 0.57 | 0.54 | **0.33** | 0.57 | 0.54 | **0.33** |
| Kin8nm | 0.14 | 0.14 | **0.11** | 0.14 | 0.14 | **0.10** |
| Naval | 0.00 | **0.00** | 0.00 | 0.00 | **0.00** | 0.00 |
| Power | **3.55** | 3.56 | 3.48 | 3.52 | 3.54 | 3.43 |
| Protein | 3.99 | 3.99 | **3.79** | 3.99 | 3.99 | **3.76** |
| Wine | 0.63 | 0.63 | **0.61** | 0.63 | 0.63 | **0.60** |
| Yacht | 0.82 | 0.84 | **0.52** | 0.83 | 0.84 | **0.50** |
| Year | **8.99** | **8.96** | **8.97** | **8.97** | **8.93** | **8.94** |

Table 2: Error and OOD detection

| Dataset | PRR | | | AUC | | |
|---------|-----|------|-----|-----|------|-----|
| | SGB | SGLB | KGB | SGB | SGLB | KGB |
| Boston | 36 | 37 | **43** | 80 | 80 | **88** |
| Concrete | 29 | 29 | **37** | 92 | 92 | **93** |
| Energy | 36 | 31 | **60** | 100 | 100 | 99 |
| Kin8nm | 18 | **19** | **20** | 45 | 45 | 41 |
| Naval | **55** | 56 | 35 | 100 | 100 | **100** |
| Power | 8 | 9 | **31** | 72 | 73 | **76** |
| Protein | 30 | 29 | **35** | 99 | 99 | **100** |
| Wine | 25 | 19 | **37** | 74 | 72 | **87** |
| Yacht | 74 | 78 | **86** | 62 | 60 | **69** |
| Year | 30 | 30 | **32** | 67 | 57 | **71** |

We note that in our setup, we *cannot* compute likelihoods as kernel $\mathcal{K}$ is defined implicitly, and its evaluation requires summing up among all possible trees structures number of which grows as $(nd)^m$ which is unfeasible, not to mention the requirement to inverse the kernel which requires $\mathcal{O}(N^{2+\omega})$ operations which additionally rules out the applicability of classical Gaussian Processes methods with our kernel. Therefore, a typical Bayesian setup is not applicable, and we resort to the uncertainty estimation setup described in Malinin et al. (2021). Also, the intractability of the kernel does not allow us to treat $\sigma, \delta$ in a fully Bayesian way, as it will require estimating the likelihood. Therefore, we fix them as constants, but we note that this will not affect the evaluation metrics for our setup as they are scale and translation invariant.

First, we compare KGB with SGLB since they both sample from similar posterior distributions. Thus, this comparison allows us to find out which of the algorithms does a better sampling from the posterior and thus provides us with more reliable estimates of knowledge uncertainty. Moreover, we consider the SGB approach as the most "straightforward" way to generate an ensemble of models.

In Table 1, we compare the predictive performance of the methods. Interestingly, we obtain improvements on almost all the datasets. Here we perform cross-validation to estimate statistical significance with paired $t$-test and highlight the approaches that are insignificantly different from the best one (p-value > 0.05). Then, we check whether uncertainty measured as the variance of the model's predictions can be used to detect errors and out-of-domain inputs. Detecting errors can be evaluated via the Prediction-Rejection Ratio (PRR) (Malinin, 2019; Malinin et al., 2020). PRR measures how well uncertainty estimates correlate with errors and rank-order them. Out-of-domain (OOD) detection is usually assessed via the area under the ROC curve (AUC-ROC) for the binary task of detecting whether a sample is OOD (Hendrycks & Gimpel, 2017). For this, one needs an OOD test set. We use the same OOD test sets (sampled from other datasets) as Malinin et al. (2021). The results of this experiment are given in Table 2. We can see that the proposed method significantly outperforms the baselines for out-of-domain detection. These improvements can be explained by the theoretical soundness of KGB: convergence properties are theoretically grounded and non-asymptotic. In contrast, for SGB, there are no general results applicable in our setting, while for SGLB the guarantees are asymptotic. In summary, these results show that our approach is superior to SGB and SGLB, achieving smaller values of RMSE and having better knowledge uncertainty estimates.

## 6 CONCLUSION

This paper theoretically analyses the classic gradient boosting algorithm. In particular, we show that GBDT converges to the solution of a certain Kernel Ridge Regression problem. We also introduce a simple modification of the classic algorithm allowing one to sample from the Gaussian posterior. The proposed method gives much better knowledge uncertainty estimates than the existing approaches.

We highlight the following important directions for future research. First, to explore how one can control the kernel and use it for better knowledge uncertainty estimates. Also, we do not analyze generalization in the current work, which is another important research topic. Finally, we need to establish universal approximation property which further justifies need for functional formalism.

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

## A    NOTATION USED IN THE PAPER

For convenience, let us list some frequently used notation:

- $X \subset \mathbb{R}^d$ — feature space;
- $d$ — dimension of feature vectors;
- $Y \subset \mathbb{R}$ — target space;
- $\rho$ — distribution of features;
- $N$ — number of samples;
- $\mathbf{z} = (\mathbf{x}_N, \mathbf{y}_N)$ — dataset;
- $\mathcal{V}$ — set of all possible tree structures;
- $L_\nu : \mathcal{V} \to \mathbb{N}$ — number of leaves for $\nu \in \mathcal{V}$;
- $D(\nu, r)$ — score used to choose a split (3);
- $\mathcal{S}$ — indices of all possible splits;
- $n$ — number of borders in our implementation of SampleTree;
- $m$ — depth of the tree in our implementation of SampleTree;
- $\beta$ — random strength;
- $\epsilon$ — learning rate;
- $\lambda$ — regularization;
- $\mathcal{F}$ — space of all possible ensembles of trees from $\mathcal{V}$;
- $\phi_\nu : X \to \{0,1\}^{L_\nu}$ — tree structure;
- $\phi_\nu^{(j)}$ — indicator of $j$-th leaf;
- $V(f)$ — empirical error of a model $f$ relative to the best possible $f' \in \mathcal{F}$;
- $k_\nu(\cdot, \cdot)$ — single tree kernel;
- $\mathcal{K}(\cdot, \cdot)$ — stationary kernel of the gradient boosting;
- $p(\cdot | f, \beta)$ — distribution of trees, $f \in \mathcal{F}$;
- $\pi(\cdot) = \lim_{\beta \to \infty} p(\cdot | f, \beta) = p(\cdot | f_*, \beta)$ — stationary distribution of trees;
- $\sigma$ — kernel scale.

The following diagram illustrates the kernels introduced in our paper:

$$\text{Weak learner' kernel } k_{\nu_\tau} : \mathcal{X} \times \mathcal{X} \to \mathbb{R}_+ \xdashrightarrow{\text{acts as } f \mapsto k_{\nu_\tau} f(\mathbf{x}_N)} \Sigma_{\nu_\tau} : \mathcal{F} \to \mathcal{F} \xrightarrow{\text{expected value} f_\tau} \Sigma_{f_\tau} : \mathcal{F} \to \mathcal{F}$$

$$\Big\downarrow \text{expected value}$$

$$\text{Iteration kernel } \mathcal{K}_{f_\tau} : \mathcal{X} \times \mathcal{X} \to \mathbb{R}_+ \xrightarrow{\tau \to \infty} \text{Stationary Kernel } \mathcal{K} : \mathcal{X} \times \mathcal{X} \to \mathbb{R}_+$$

$$\Big\updownarrow \text{used in dot product}$$

$$\mathcal{H} = \big(\mathcal{F}, \langle \cdot, \cdot \rangle_{\mathcal{H}}\big)$$

## B    ADDITIONAL RELATED WORK

Let us briefly discuss some additional related work. Mondrian forest method (Balog et al., 2016) and Generalized Random Forests (Athey et al., 2019), besides having links to the kernel methods, in fact, have the underlying limiting RKHS space that is much smaller than the space of all possible ensembles built on the same weak learners due to the independence of the trees that are added to the ensemble. Therefore, there is an issue of high bias when comparing plain gradient boosting with the plain random forest method. Also, these two methods are built from scratch to obtain an RKHS

interpretation while we provide a link between the existing standard gradient boosting approaches to the kernel methods, i.e., we do not create a novel gradient boosting algorithm but rather show that the existing ones already have such a link to derive convergence rates and to exploit such linkage to obtain formal Gaussian process interpretation of the gradient boosting learning to get uncertainty estimates using well-established gradient boosting libraries.

Let us mention that there are approaches that study kernels induced by tree ensembles through the perspective of Neural Tangent Kernel (Kanoh & Sugiyama, 2022), though this analysis is not applicable for classical gradient boosting, while ours is.

Let us also briefly discuss the papers on Neural Tangent Kernel, e.g., Jacot et al. (2018); Li & Liang (2018); Allen-Zhu et al. (2019); Du et al. (2019), that study deep learning convergence through the perspective of kernel methods. Though such works share similarities with what we do, there are fundamental differences. First, our work is not in the over-parametrization regime, i.e., our kernel method correspondence works for tree ensembles with fixed parameters, but the correspondence is achieved as the number of iterations goes to infinity. It is worth noting that the kernel method perspective on deep learning basically establishes that each trained deep learning model is a sample from Gaussian Process posterior (Lee et al., 2020; 2018; Yang, 2019; Cho & Saul, 2009), i.e., is sample-then-optimize. For boosting, we achieve this only by introducing Algorithm 4 relying on Algorithm 3, which in its essence, is random initialization for gradient boosting. The classic gradient boosting (Algorithm 2) can be considered as the mean value of the Gaussian Process, which has no analogs in the world of deep learning, and to achieve convergence to posterior mean there, one needs to average among many trained models. This can be considered as an advantage of gradient boosting over deep learning that we derive in our paper.

## C  CONVEX OPTIMIZATION IN FUNCTIONAL SPACES

In this section, we formulate basic definitions of differentiability in functional spaces and the theorem on the convergence of gradient descent in functional spaces. For the proof of the theorem and further details on convex optimization in functional space, the reader can consult Luenberger (1969).

We consider $\mathcal{H}$ to be a Hilbert space with some scalar product $\langle \cdot, \cdot \rangle_{\mathcal{H}}$.

**Definition C.1.** We say that $F : \mathcal{H} \to \mathbb{R}$ is Fréchet differentiable if for any $f \in \mathcal{H}$ there exists a bounded linear functional $L_f : \mathcal{H} \to \mathbb{R}$ such that $\forall h \in \mathcal{H}$

$$F(f + h) = F(f) + L_f[h] + o(\|h\|).$$

The value of $L_f : \mathcal{H} \to \mathbb{R}$ is denoted by $\mathcal{D}_f F(f)$ and is called a Fréchet differential of $F$ at point $f$. So, Fréchet differential is a functional $\mathcal{D}_f F : \mathcal{H} \to \mathcal{B}(\mathcal{H}, \mathbb{R})$, where $\mathcal{B}(X, Y)$ denotes a normed space of linear bounded functionals from $X$ to $Y$.

**Definition C.2.** Let $F : \mathcal{H} \to \mathbb{R}$ be Fréchet differentiable with a Fréchet differential $\mathcal{D}_f F(f)$ that is a bounded linear functional. Then, by the Riesz theorem there exists a unique $h_f$ such that $\big(\mathcal{D}_f F(f)\big)[h] = \langle h_f, h \rangle_{\mathcal{H}} \ \forall h \in \mathcal{H}$. We call such element a *gradient* of $F$ in $\mathcal{H}$ at $f \in \mathcal{H}$ and denote it by $\nabla_{\mathcal{H}} F(f) = h_f \in \mathcal{H}$.

**Definition C.3.** $F : \mathcal{H} \to \mathbb{R}$ is said to be twice Fréchet differentiable if $\mathcal{D}_f F$ is Fréchet differentiable, where the definition of Fréchet differential is analogous to Definition C.1 with the only difference that $\mathcal{D}_f F$ takes values in $\mathcal{B}(\mathcal{H}, \mathbb{R})$. The second Fréchet differential is denoted by $\mathcal{D}_f^2 F : \mathcal{H} \to \mathcal{B}(\mathcal{H}, \mathcal{B}(\mathcal{H}, \mathbb{R}))$. As there is an isomorphism between $\mathcal{B}(\mathcal{H}, \mathcal{B}(\mathcal{H}, \mathbb{R}))$ and $\mathcal{B}(\mathcal{H} \times \mathcal{H}, \mathbb{R})$, we can consider the second Fréchet differential to take values in $\mathcal{B}(\mathcal{H} \times \mathcal{H}, \mathbb{R})$. Henceforth, we will not differentiate between $\mathcal{B}(\mathcal{H}, \mathcal{B}(\mathcal{H}, \mathbb{R}))$ and $\mathcal{B}(\mathcal{H} \times \mathcal{H}, \mathbb{R})$.

**Definition C.4.** A linear operator $P : \mathcal{H} \times \mathcal{H} \to \mathbb{R}$ is said to be semi-positive definite (denoted by $P \succeq 0$) if $\forall f \in \mathcal{H}$ we have $P(f, f) \geq 0$. $P$ is said to be positive definite ($P \succ 0$) if $\forall f \in \mathcal{H} \setminus \{0\}$ we have $P(f, f) > 0$.

**Definition C.5.** Given two linear operators $P, G : \mathcal{H} \times \mathcal{H} \to \mathbb{R}$ we write $P \succeq G$ if $P - G \succeq 0$ and $P \succ G$ if $P - G \succ 0$.

Let $I \in B(\mathcal{H} \times \mathcal{H}, \mathbb{R})$ be a linear operator defined as $I(g, h) = (g, h)_{\mathcal{H}}$.

**Theorem C.6.** *Let $F$ be bounded below and twice Fréchet differentiable functional on a Hilbert space $\mathcal{H}$. Assume that $\mathcal{D}_f^2 F(f)$ satisfies $0 \prec mI \preceq \mathcal{D}_f^2 F(f) \preceq \mu I$. Then the gradient descent*

*scheme:*

$$f_{k+1} = f_k - \epsilon \nabla_{\mathcal{H}} F(f_k)$$

*converges to $f_*$ that minimizes F.*

*Proof.* For the proof see Luenberger (1969). □

## D  KERNEL RIDGE REGRESSION AND RKHS

**Definition D.1.** $\mathcal{K} : X \times X \to \mathbb{R}$ is called a kernel function if it is positive semi-definite, i.e., $\forall N \in \mathbb{N}^+ \ \forall \mathbf{x}_N \in \mathcal{X}^N : \ \mathcal{K}(\mathbf{x}_N, \mathbf{x}_N) \succeq 0$.

**Definition D.2.** For any kernel function we can define a Reproducing Kernel Hilbert Space (RKHS)

$$\mathcal{H}(\mathcal{K}) = \text{span}\left\{\mathcal{K}(\cdot, x) \big| x \in X\right\}$$

with a scalar product such that

$$\langle f, \mathcal{K}(\cdot, x)\rangle_{\mathcal{H}(\mathcal{K})} = f(x).$$

Consider the following Kernel Ridge Regression problem:

$$V(f, \lambda) = \frac{1}{2N}\|\mathbf{y}_N - f(\mathbf{x}_N)\|^2_{\mathbb{R}^N} + \frac{\lambda}{2N}\|f\|^2_{\mathcal{H}(K)} - \min_{f \in \mathcal{H}(\mathcal{K})} V(f, \lambda) \to \min_{f \in \mathcal{H}(\mathcal{K})}$$

and the following Kernel Ridgeless Regression problem:

$$V(f) = \frac{1}{2N}\|\mathbf{y}_N - f(\mathbf{x}_N)\|^2_{\mathbb{R}^N} - \min_{f \in \mathcal{H}(\mathcal{K})} V(f) \to \min_{f \in \mathcal{H}(\mathcal{K})}.$$

**Lemma D.3.** $\min_{\mathcal{H}(\mathcal{K})} V(f, \lambda)$ *has the only solution*

$$f_*^\lambda = \mathcal{K}(\cdot, \mathbf{x}_N)(\mathcal{K}(\mathbf{x}_N, \mathbf{x}_N) + \lambda I)^{-1}\mathbf{y}_N.$$

*Proof.* First, let us show that $f_*^\lambda \in \text{span}\left\{\mathcal{K}(\cdot, x_i)\right\}$. Let $\mathcal{H}(\mathcal{K}) = \text{span}\left\{\mathcal{K}(\cdot, x_i)\right\} \oplus \text{span}\left\{\mathcal{K}(\cdot, x_i)\right\}^\perp$ and consider the projector $P : \mathcal{H}(\mathcal{K}) \to \mathcal{H}(\mathcal{K})$ onto the space $\text{span}\left\{\mathcal{K}(\cdot, x_i)\right\}$.

It is easy to show that $P(f)(\mathbf{x}_N) = f(\mathbf{x}_N)$ for any $f \in \mathcal{H}(\mathcal{K})$. Indeed,

$$(f - P(f))[\mathbf{x}_N] = \langle f - P(f), \mathcal{K}(\cdot, \mathbf{x}_N)\rangle = 0.$$

If $f_*^\lambda$ does not lie in $\text{span}\left\{\mathcal{K}(\cdot, x_i)\right\}$, then $\|f_*^\lambda\|_{\mathcal{H}(\mathcal{K})} > \|P(f_*^\lambda)\|_{\mathcal{H}(\mathcal{K})}$ and $V(P(f_*^\lambda), \lambda) < V(f_*^\lambda, \lambda)$. We get a contradiction with the minimality of $f_*^\lambda$.

Now, let us prove the existence of $f_*^\lambda$. Consider $f = \mathcal{K}(\cdot, \mathbf{x}_N)c$, where $c \in \mathbb{R}^N$. Then, we find the optimal $c$ by taking a derivative of $V(f, \lambda)$ with respect to $c$ and equating it to zero:

$$\mathcal{K}(\mathbf{x}_N, \mathbf{x}_N)(\mathcal{K}(\mathbf{x}_N, \mathbf{x}_N)c - y_N) + \lambda \mathcal{K}(\mathbf{x}_N, \mathbf{x}_N)c = 0.$$

Then,

$$c_v = (\mathcal{K}(\mathbf{x}_N, \mathbf{x}_N) + \lambda I)^{-1}(y_N + v),$$

where $v \in \ker \mathcal{K}(\mathbf{x}_N, \mathbf{x}_N)$. Note that all $\mathcal{K}(\cdot, \mathbf{x}_N)c_v$ are equal. Then, we have the only solution of the KRR problem:

$$f_*^\lambda = \mathcal{K}(\cdot, \mathbf{x}_N)(\mathcal{K}(\mathbf{x}_N, \mathbf{x}_N) + \lambda I)^{-1}\mathbf{y}_N.$$

□

**Lemma D.4.** $\min_{\mathcal{H}(\mathcal{K})} V(f)$ *has the only solution in* $\text{span}\left\{\mathcal{K}(\cdot, \mathbf{x}_i)\right\}$ *and it is the solution of minimum RKHS norm:*

$$f_* = \mathcal{K}(\cdot, \mathbf{x}_N)\mathcal{K}(\mathbf{x}_N, \mathbf{x}_N)^\dagger \mathbf{y}_N.$$

*Proof.* Consider $f = \mathcal{K}(\cdot, \mathbf{x}_N)c$, where $c \in \mathbb{R}^N$. Now consider $V(f)$ and differentiate it with respect to $c$. If we equate the derivative to zero, we get:

$$\frac{1}{N}\mathcal{K}(\mathbf{x}_N, \mathbf{x}_N)(\mathcal{K}(\mathbf{x}_N, \mathbf{x}_N)c - \mathbf{y}_N) = 0\,.$$

Then, $\mathcal{K}(\mathbf{x}_N, \mathbf{x}_N)c - (\mathbf{y}_N + v) = 0$ for some $v \in \ker \mathcal{K}(\mathbf{x}_N, \mathbf{x}_N)$. Note that $\mathbf{y}_N + \ker \mathcal{K}(\mathbf{x}_N, \mathbf{x}_N) \cap \operatorname{Im} \mathcal{K}(\mathbf{x}_N, \mathbf{x}_N) \neq \emptyset$. Then, for any $v$ such that $\mathbf{y}_N + v \in \operatorname{Im} \mathcal{K}(\mathbf{x}_N, \mathbf{x}_N)$ there exists a solution $c_v = \mathcal{K}(\mathbf{x}_N, \mathbf{x}_N)^\dagger(\mathbf{y}_N + v) + \ker \mathcal{K}(\mathbf{x}_N, \mathbf{x}_N)$. This follows from the fact that $\mathcal{K}(\mathbf{x}_N, \mathbf{x}_N)\mathcal{K}(\mathbf{x}_N, \mathbf{x}_N)^\dagger$ is an orthoprojector onto $\operatorname{Im} \mathcal{K}(\mathbf{x}_N, \mathbf{x}_N)$. Note that

$$f_* = \mathcal{K}(\cdot, \mathbf{x}_N)(\mathcal{K}(\mathbf{x}_N, \mathbf{x}_N)^\dagger(\mathbf{y}_N + v) + \ker \mathcal{K}(\mathbf{x}_N, \mathbf{x}_N)) = \mathcal{K}(\cdot, \mathbf{x}_N)\mathcal{K}(\mathbf{x}_N, \mathbf{x}_N)^\dagger \mathbf{y}_N.$$

Then, the existence and uniqueness of $f_*$ follow. $\qquad\square$

Now, consider a linear space $\mathcal{F} \subset L_2(\rho)$ of all possible ensembles of trees from $\mathcal{V}$:

$$\mathcal{F} = \operatorname{span}\left\{\phi_\nu^{(j)}(\cdot) : \mathcal{X} \to \{0, 1\} \big| \nu \in \mathcal{V}, j \in \{1, \ldots, L_\nu\}\right\}.$$

Define the unique function:

$$\{f_*\} = \lim_{\lambda \to 0_+} \operatorname*{arg\,min}_{f \in \mathcal{F}} V(f, \lambda) \subset \operatorname*{arg\,min}_{f \in \mathcal{F}} V(f).$$

Then following two lemmas hold:

**Lemma D.5.** $\langle \mathbf{y}_N - f_*(\mathbf{x}_N), f(\mathbf{x}_N)\rangle_{\mathbb{R}^N} = 0$ *for any* $f \in \mathcal{F}$.

*Proof.* Assume that $\langle \mathbf{y}_N - f_*(\mathbf{x}_N), f(\mathbf{x}_N)\rangle_{\mathbb{R}^N} \neq 0$ for some $f \in \mathcal{F}$. Then, for some $f \in \mathcal{F}$, $\langle \mathbf{y}_N - f_*(\mathbf{x}_N), f(\mathbf{x}_N)\rangle_{\mathbb{R}^N} > 0$. We have:

$$\|\mathbf{y}_N - (f_* + \alpha f)(\mathbf{x}_N)\|_{\mathbb{R}^N}^2$$
$$= \|\mathbf{y}_N - f_*(\mathbf{x}_N)\|_{\mathbb{R}^N}^2 - 2\alpha\langle \mathbf{y}_N - f_*(\mathbf{x}_N), f(\mathbf{x}_N)\rangle_{\mathbb{R}^N} + \alpha^2\|f(\mathbf{x}_N)\|_{\mathbb{R}^N}^2$$
$$< \|\mathbf{y}_N - f_*(\mathbf{x}_N)\|_{\mathbb{R}^N}^2$$

for small enough $\alpha > 0$, which contradicts with the definition of $f_*$.

$\qquad\square$

**Lemma D.6.**
$$V(f, \lambda) = \frac{1}{2N}\|f_*(\mathbf{x}_N) - f(\mathbf{x}_N)\|_{\mathbb{R}^N}^2 + \frac{\lambda}{2N}\|f\|_{\mathcal{H}}^2 - C_\lambda$$
*where* $C_\lambda = \inf_{f \in F} L(f, \lambda) - \inf_{f \in F} L(f) \geq 0$.

*Proof.* We need to prove it only for $V(f)$ without regularization as for regularized it follows immediately (note that $f_*$ is minimizer of not regularized objective). By definition,

$$V(f) = \frac{1}{2N}\|\mathbf{y}_N - f(\mathbf{x}_N)\|_{\mathbb{R}^N}^2 - \frac{1}{2N}\|\mathbf{y}_N - f_*(\mathbf{x}_N)\|_{\mathbb{R}^N}^2\,.$$

Now, let us prove that

$$\|\mathbf{y}_N - f(\mathbf{x}_N)\|_{\mathbb{R}^N}^2 - \|\mathbf{y}_N - f_*(\mathbf{x}_N)\|_{\mathbb{R}^N}^2 - \|f_*(\mathbf{x}_N) - f(\mathbf{x}_N)\|_{\mathbb{R}^N}^2 = 0\,.$$

Indeed,

$$\|\mathbf{y}_N - f(\mathbf{x}_N)\|_{\mathbb{R}^N}^2 - \|\mathbf{y}_N - f_*(\mathbf{x}_N)\|_{\mathbb{R}^N}^2 - \|f_*(\mathbf{x}_N) - f(\mathbf{x}_N)\|_{\mathbb{R}^N}^2$$
$$= -2\langle f_*(\mathbf{x}_N), f_*(\mathbf{x}_N)\rangle_{\mathbb{R}^N} - 2\langle \mathbf{y}_N, f(\mathbf{x}_N) - f_*(\mathbf{x}_N)\rangle_{\mathbb{R}^N} + 2\langle f_*(\mathbf{x}_N), f(\mathbf{x}_N)\rangle_{\mathbb{R}^N}$$
$$= -2\langle \mathbf{y}_N - f_*(\mathbf{x}_N), f(\mathbf{x}_N) - f_*(\mathbf{x}_N)\rangle_{\mathbb{R}^N} = 0\,,$$

where the last equality follows from the previous lemma. $\qquad\square$

**Lemma D.7.** $V(f, \lambda) = \frac{1}{2N}\|f_*^\lambda(\mathbf{x}_N) - f(\mathbf{x}_N)\|_{\mathbb{R}^N}^2 + \frac{\lambda}{2N}\|f_*^\lambda - f\|_{\mathcal{H}}^2$.

*Proof.* $f_*^\lambda$ is optimum for $V(f, \lambda)$. Then Fréchet derivative at $f_*^\lambda$ equals 0:

$$\mathcal{D}_f V(f_*^\lambda, \lambda) = 0.$$

Consider then writing:

$$V(f, \lambda) = V(f_*^\lambda, \lambda) + \mathcal{D}_f V(f_*^\lambda, \lambda)[f - f_*^\lambda] + \frac{1}{2}\mathcal{D}_f^2 V(f_*^\lambda, \lambda)[f - f_*^\lambda, f - f_*^\lambda]$$

$$= \frac{1}{2N}\|f_*^\lambda(\mathbf{x}_N) - f(\mathbf{x}_N)\|_{\mathbb{R}^N}^2 + \frac{\lambda}{2N}\|f_*^\lambda(\mathbf{x}_N) - f(\mathbf{x}_N)\|_{\mathbb{R}^N}^2.$$

The explicit formula for the Fréchet Derivative of $V(f, \lambda)$ can be found in Appendix E. $\qquad\square$

## E   GAUSSIAN PROCESS INFERENCE

In this section, we prove Lemma 2.1 from Section 2.3 of the main text.

Firstly, consider the following regularized error functional:

$$V(f, \lambda) = \frac{1}{2N}\sum_{i=1}^N \left(f(x_i) - y_i\right)^2 + \frac{\lambda}{2N}\|f\|_\mathcal{H}^2 - \min_{f \in \mathcal{H}(\mathcal{K})}\left(\frac{1}{2N}\sum_{i=1}^N \left(f(x_i) - y_i\right)^2 + \frac{\lambda}{2N}\|f\|_\mathcal{H}^2\right).$$

With this functional we can consider the following optimization problem:

$$\min_{f \in \mathcal{H}(\mathcal{K})} V(f, \lambda),$$

which is called as Kernel Ridge Regression.

We will show that this functional satisfies the conditions needed for Theorem C.6. We will also deduce the formula of the gradient of $V$ in order to show that gradient descent takes the form (2).

**Lemma E.1.** *$V(f, \lambda)$ is Fréchet differentiable with the differential given by:*

$$\mathcal{D}_f V(f, \lambda) = \frac{\lambda}{N}\langle f, \cdot\rangle_{\mathcal{H}(\mathcal{K})} + \frac{1}{N}\sum_{i=1}^N \left(f(x_i) - y_i\right)ev_{x_i},$$

*where $ev_{x_i} : \mathcal{H}(\mathcal{K}) \to \mathbb{R}$ is a bounded linear functional such that $ev_{x_i}(f) = f(x_i) = (f, \mathcal{K}(x_i, \cdot))_{\mathcal{H}(\mathcal{K})}$.*[8]

*Proof.* As Fréchet differential is linear, we only need to find Fréchet differential for $(f(x_i) - y_i)^2$.

Note that $(f(x_i) - y_i)^2$ is a composition of two functions:

$$F : \mathcal{H}(\mathcal{K}) \to \mathbb{R}, \ F = ev_{x_i} - y_i,$$

$$G : \mathbb{R} \to \mathbb{R}, \ G(x) = x^2.$$

The differential of the composition can be found as:

$$\mathcal{D}_f G(F(f)) = \frac{\partial}{\partial x}G(F(f))\mathcal{D}_f F(f),$$

$$\mathcal{D}_f G(F(f)) = 2(f(x_i) - y_i)ev_{x_i},$$

where $\mathcal{D}_f F(f) = ev_{x_i}$ because

$$ev_{x_i}(f + h) - y_i = ev_{x_i}(f) - y_i + ev_{x_i}(h).$$

$\qquad\square$

**Lemma E.2.** *The gradient of $V(f, \lambda)$, Riesz representative of the functional above, is given by:*

$$\nabla_f V(f, \lambda) = \frac{\lambda}{N}f + \frac{1}{N}\sum_{i=1}^N (f(x_i) - y_i)\mathcal{K}_{x_i}.$$

---

[8]We further use the notation $\mathcal{K}_{x_i} := \mathcal{K}(\cdot, x_i)$.

*Proof.* Follows from the previous lemma. □

**Lemma E.3.** $V(f, \lambda)$ *is twice Fréchet differentiable with the differential given by:*

$$\mathcal{D}_f^2 V : \mathcal{H}(\mathcal{K}) \to B(\mathcal{H}(\mathcal{K}), B(\mathcal{H}(\mathcal{K}), \mathbb{R})),$$

$$\mathcal{D}_f^2 V(f, \lambda)[h] = \frac{\lambda}{N} \langle h, \cdot \rangle_{\mathcal{H}(\mathcal{K})} + \frac{1}{N} \sum_{i=1}^N h(x_i) ev_{x_i}.$$

*Proof.* Due to the linearity of Fréchet differential and lemma E.1 we need to find only Fréchet differential for $(f(x_i) - y_i)ev_{x_i}$.

Consider $S(f) = (f(x_i) - y_i)ev_{x_i}$. Then we need to find $V_f \in B(\mathcal{H}(\mathcal{K}), B(\mathcal{H}(\mathcal{K}), \mathbb{R}))$ such that $S(f + h) = S(f) + V_f[h] + o(\|h\|)$.

It is easy to show that $h \mapsto h(x_i)ev_{x_i} \in B(\mathcal{H}(\mathcal{K}), B(\mathcal{H}(\mathcal{K}), \mathbb{R}))$ and $S(f+h) = S(f) + h(x_i)ev_{x_i}$. Thus, we get that $\mathcal{D}_f S(f)[h] = h(x_i)ev_{x_i}$. From this the statement of the lemma follows. □

Given all the above lemmas, as a corollary of Theorem C.6, we have the following.

**Corollary E.4.** *Gradient descent, defined by the following iterative scheme*

$$f_{\tau+1} = \left(1 - \frac{\lambda\epsilon}{N}\right) f_\tau - \epsilon \frac{1}{N} \sum_{i=1}^N (f_\tau(x_i) - y_i) \mathcal{K}_{x_i}, \quad f_0 = \mathbb{0}_{L_2(\rho)}$$

*converges to the optimum of $V(f, \lambda)$. Thus,*

$$f_*^\lambda = \lim_{\tau \to \infty} f_\tau = \mathcal{K}(\cdot, \mathbf{x}_N) \left( \mathcal{K}(\mathbf{x}_N, \mathbf{x}_N) + \lambda I_N \right)^{-1} \mathbf{y}_N. \tag{6}$$

*Proof.* By Lemma E.2, our update rule has the form

$$f_{\tau+1} = f_\tau - \epsilon \nabla_{\mathcal{H}} V(f_\tau, \lambda).$$

Then, we will find $m, \mu$ such that $0 \prec mI \preceq \mathcal{D}_f^2 V(f, \lambda) \preceq \mu I$. By Lemma E.3,

$$\mathcal{D}_f^2 V(f, \lambda)[g, h] = \frac{\lambda}{N} \langle g, h \rangle_{\mathcal{H}(\mathcal{K})} + \frac{1}{N} g(\mathbf{x}_N)^T h(\mathbf{x}_N).$$

Then, we can take $m = \frac{\lambda}{N}$. Let us also write

$$\mathcal{D}_f^2 V(f, \lambda)[g, g] = \frac{\lambda}{N} \|g\|_{\mathcal{H}(\mathcal{K})}^2 + \frac{1}{N} \|g(\mathbf{x}_N)\|^2 =$$

$$\frac{\lambda}{N} \|g\|_{\mathcal{H}(\mathcal{K})}^2 + \frac{1}{N} \|\langle g, \mathcal{K}(\cdot, \mathbf{x}_N) \rangle_{\mathcal{H}(\mathcal{K})}\|^2 \le \left( \frac{\lambda}{N} + \frac{1}{N} \max_{x \in \mathcal{X}} \mathcal{K}(x, x) \right) \|g\|_{\mathcal{H}(\mathcal{K})}^2.$$

Then, we can take $\mu = \frac{\lambda}{N} + \frac{1}{N} \max_{x \in \mathcal{X}} \mathcal{K}(x, x)$. By theorem C.6 and lemma D.3 the corollary follows.

□

**Lemma E.5.** *Consider the gradient descent:*

$$f_{\tau+1} = \left(1 - \frac{\lambda\epsilon}{N}\right) f_\tau - \epsilon \frac{1}{N} \sum_{i=1}^N (f_\tau(x_i) - y_i) \mathcal{K}_{x_i},$$

$$f_0 = \mathbb{0}_{L_2(\rho)},$$

$$f_\infty = \lim_{\tau \to \infty} f_\tau$$

*and the following randomization scheme:*

*1. sample $f^{init} \sim \mathcal{GP}(\mathbb{0}_{L_2(\rho)}, \sigma^2 \mathcal{K} + \delta^2 \mathrm{Id}_{L_2})$;*

*2. set new labels $\mathbf{y}_N^{new} = \mathbf{y}_N - f^{init}(\mathbf{x}_N)$;*

*3. fit GD $f_\tau(\cdot)$ on $\mathbf{y}_N^{new}$ assuming $f_0(\cdot) = \mathbb{0}_{L_2(\rho)}$;*

*4. output $\hat{f}(\cdot) = f^{init}(\cdot) + f_\infty(\cdot)$ as final model.*

*Then, $\hat{f}$ from the scheme above follows the Gaussian Process posterior with the following mean and covariance:*

$$\mathbb{E}\hat{f}(x) = \mathcal{K}(x, \mathbf{x}_N)\Big(\mathcal{K}(\mathbf{x}_N, \mathbf{x}_N) + \lambda I_N\Big)^{-1}\mathbf{y}_N,$$

$$\mathrm{cov}(\hat{f}(x)) = \delta^2 + \sigma^2\Big(\mathcal{K}(x, x) - \mathcal{K}(x, \mathbf{x}_N)\Big(\mathcal{K}(\mathbf{x}_N, \mathbf{x}_N) + \lambda I_N\Big)^{-1}\mathcal{K}(\mathbf{x}_N, x)\Big).$$

*Proof.*

$$f_\infty = \mathcal{K}(\cdot, \mathbf{x}_N)\Big(\mathcal{K}(\mathbf{x}_N, \mathbf{x}_N) + \lambda I_N\Big)^{-1}\mathbf{y}_N^{new} = \mathcal{K}(\cdot, \mathbf{x}_N)\Big(\mathcal{K}(\mathbf{x}_N, \mathbf{x}_N) + \lambda I_N\Big)^{-1}(\mathbf{y}_N - f^{init}(\mathbf{x}_N)).$$

Let us find the distribution of $\hat{f}$ at $x \in \mathbb{R}^n$. It can be easily seen that:

$$\mathbb{E}\hat{f}(x) = \mathcal{K}(x, \mathbf{x}_N)\Big(\mathcal{K}(\mathbf{x}_N, \mathbf{x}_N) + \lambda I_N\Big)^{-1}\mathbf{y}_N.$$

Let us now calculate covariance:

$$\begin{aligned}
\mathrm{cov}\hat{f}(x) &= \mathbb{E}(\hat{f}(x) - \mathbb{E}\hat{f}(x))(\hat{f}(x) - \mathbb{E}\hat{f}(x))^T \\
&= \mathbb{E}\big(f^{init}(x) - \mathcal{K}(x, \mathbf{x}_N)\Big(\mathcal{K}(\mathbf{x}_N, \mathbf{x}_N) + \lambda I_N\Big)^{-1}f^{init}(\mathbf{x}_N)\big) \\
&\qquad \cdot \big(f^{init}(x) - \mathcal{K}(x, \mathbf{x}_N)\Big(\mathcal{K}(\mathbf{x}_N, \mathbf{x}_N) + \lambda I_N\Big)^{-1}f^{init}(\mathbf{x}_N)\big)^T \\
&= \mathbb{E}f^{init}(x)f^{init}(x)^T - \mathbb{E}f^{init}(x)f^{init}(\mathbf{x}_N)^T\Big(\mathcal{K}(\mathbf{x}_N, \mathbf{x}_N) + \lambda I_N\Big)^{-1}\mathcal{K}(\mathbf{x}_N, x) \\
&\qquad - \mathcal{K}(x, \mathbf{x}_N)\Big(\mathcal{K}(\mathbf{x}_N, \mathbf{x}_N) + \lambda I_N\Big)^{-1}\mathbb{E}f^{init}(\mathbf{x}_N)f^{init}(x)^T \\
&\quad + \mathcal{K}(x, \mathbf{x}_N)\Big(\mathcal{K}(\mathbf{x}_N, \mathbf{x}_N) + \lambda I_N\Big)^{-1}\mathbb{E}f^{init}(\mathbf{x}_N)f^{init}(\mathbf{x}_N)^T\Big(\mathcal{K}(\mathbf{x}_N, \mathbf{x}_N) + \lambda I_N\Big)^{-1}\mathcal{K}(\mathbf{x}_N, x) \\
&= \delta^2 + \sigma^2\big(\mathcal{K}(x, x) - 2\mathcal{K}(x, \mathbf{x}_N)(\mathcal{K}(\mathbf{x}_N, \mathbf{x}_N) + \lambda I_N)^{-1}\mathcal{K}(\mathbf{x}_N, x)\big) \\
&\qquad + \sigma^2\mathcal{K}(x, \mathbf{x}_N)(\mathcal{K}(\mathbf{x}_N, \mathbf{x}_N) + \lambda I_N)^{-1}\mathcal{K}(\mathbf{x}_N, x)\big) \\
&= \delta^2 + \sigma^2\big(\mathcal{K}(x, x) - \mathcal{K}(x, \mathbf{x}_N)(\mathcal{K}(\mathbf{x}_N, \mathbf{x}_N) + \lambda I_N)^{-1}\mathcal{K}(\mathbf{x}_N, x)\big),
\end{aligned}$$

which is exactly what we need. $\qquad\square$

## F  DISTRIBUTION OF TREES

**Lemma F.1** (Lemma 3.2 in the main text).

$$p(\nu | f, \beta) = \sum_{\varsigma \in \mathcal{P}_m} \prod_{i=1}^m \frac{e^{\frac{D(\nu_{\varsigma, i}, r)}{\beta}}}{\sum_{s \in \mathcal{S} \setminus \nu_{\varsigma, i-1}} e^{\frac{D((\nu_{\varsigma, i-1}, s), r)}{\beta}}},$$

*where the sum is over all permutations $\varsigma \in \mathcal{P}_m$, $\nu_{\varsigma, i} = (s_{\varsigma(1)}, \ldots, s_{\varsigma(i)})$, and $\nu = (s_1, \ldots, s_m)$.*

*Proof.* Let us fix some permutation $\varsigma \in \mathcal{P}_m$. W.l.o.g., let $\varsigma = \mathrm{id}_{\mathcal{P}_m}$, i.e. $\varsigma(i) = i \, \forall i$. It remains to derive the formula for the fixed permutation. The probability of adding the next split given the previously build tree is:

$$P(\nu_{i-1} \cup s_i | \nu_{i-1}) = \frac{e^{\frac{1}{\beta}D(\nu_i, r)}}{\sum_{s \in \mathcal{S} \setminus \nu_{i-1}} e^{\frac{1}{\beta}D((\nu_{i-1}, s), r)}},$$

which comes from (4) and the Gumbel-SoftMax trick. Then, we decompose the probability $P(\nu)$ of a tree as:

$$P(\nu) = \prod_{i=1}^m P(\nu_{i-1} \cup s_i | \nu_{i-1}),$$

and so for the fixed permutation we have

$$P(\nu) = \prod_{i=1}^{m} \frac{e^{\frac{1}{\beta}D(\nu_i, r)}}{\sum_{s \in \mathcal{S} \setminus \nu_{i-1}} e^{\frac{1}{\beta}D((\nu_{i-1}, s), r)}} \, .$$

Then we sum over all permutations and the lemma follows.

$\square$

Now, let us define the following value indicating how different are the distribution of trees for $f$ and $f_*$:

$$\Gamma_\beta(f) = \max \left\{ \max_{\nu \in \mathcal{V}} \left| \frac{p(\nu|f_*, \beta)}{p(\nu|f, \beta)} \right|, \max_{\nu \in \mathcal{V}} \left| \frac{p(\nu|f, \beta)}{p(\nu|f_*, \beta)} \right| \right\} \, .$$

**Lemma F.2.** *The following bound relates the distributions.*

$$\Gamma_\beta(f) \le e^{\frac{2mV(f)}{\beta}} \, .$$

*Proof.* Consider $\pi = p(\cdot|f_*, \beta)$ and the following expression $P(\nu, \varsigma)$:

$$P(\nu, \varsigma) := \prod_{i=1}^{m} \frac{e^{\frac{1}{\beta}D(\nu_{\varsigma, i}, r)}}{\sum_{s \in \mathcal{S} \setminus \nu_{\varsigma, i-1}} e^{\frac{1}{\beta}D((\nu_{\varsigma, i-1}, s), r)}} \, .$$

Then,

$$\sum_{\varsigma \in \mathcal{P}_m} P(\nu, \varsigma) \le \sum_{\varsigma \in \mathcal{P}_m} e^{\frac{m}{\beta}D(\nu, r)} \prod_{i=1}^{m} \frac{1}{\sum_{s \in \mathcal{S} \setminus \nu_{\varsigma, i-1}} 1} \le e^{\frac{2mV(f)}{\beta}} \pi(\nu) \, .$$

where in second inequality we used $D(\nu, r) \le 2V(f)$ which straightly follows from the definition.

By noting that the probabilities remain the same if we shift $D(\cdot, r) \leftarrow D(\cdot, r) - 2V(f)$ which becomes everywhere non-positive and allows us to do the above trick once more but in reverse manner: if we formally replace the $D$ with such modified function and repeat the steps with reversing the inequalities which is needed since the new function is everywhere negative then the lemma follows.

$$\sum_{\varsigma \in \mathcal{P}_m} P(\nu, \varsigma) \ge \sum_{\varsigma \in \mathcal{P}_m} e^{\sum_{i=1}^{m} \frac{1}{\beta}D(\nu_{\varsigma, i}, r) - \frac{m}{\beta}2V(f)} \prod_{i=1}^{m} \frac{1}{\sum_{s \in \mathcal{S} \setminus \nu_{i-1}} 1} \ge e^{-\frac{2mV(f)}{\beta}} \pi(\nu) \, .$$

$\square$

## G    PROOF OF THEOREM 3.8

### G.1    RKHS STRUCTURE

In section 3.4 we defined RKHS structure on $F$ as:

$$\langle f, \mathcal{K}(\cdot, x) \rangle_{\mathcal{H}(\mathcal{K})} = f(x)$$

and we introduced the kernels $k_\nu, \mathcal{K}_f, \mathcal{K}_\pi$. Let us also define a kernel $\mathcal{K}_p(\cdot, \cdot) = \sum_{\nu \in \mathcal{V}} k_\nu(\cdot, \cdot) p(\nu)$ for arbitrary distribution $p$ on $\mathcal{V}$. This way, taking $p$ as $\delta_\nu(\cdot)$, $p(\nu \mid f, \beta)$, $\pi(\cdot)$ we get $\mathcal{K}_p$ equal to $k_\nu, \mathcal{K}_f, \mathcal{K}$, respectively.

For each kernel, we define the operator associated with it denoted similarly:

$$\mathcal{K}_p : \mathcal{F} \to \mathcal{F},$$

$$f \mapsto \int_X \mathcal{K}_p(\cdot, x) f(x) \rho(\mathrm{d}x).$$

**Lemma G.1.** *Consider two positive semidefinite operators on a finite dimensional vector space $V$: $A : V \to V$ and $B : V \to V$ such that $A \succeq B$. Then, $\mathrm{Im}\, A \ge \mathrm{Im}\, B$.*

**Lemma G.2.** $\mathcal{K}_p : \mathcal{F} \to \mathcal{F}$ *is invertible for $p$ non-vanishing on $\mathcal{V}$.*

*Proof.* Note that $\operatorname{Im} k_\nu = \operatorname{span}\{\phi_\nu^j \mid j = 1, \ldots, L_\nu\}$ and $p(\nu)k_\nu \preceq \mathcal{K}_p$. Then, $\operatorname{Im}\mathcal{K}_p = \mathcal{F}$ follows from lemma G.1. Thus, $\mathcal{K}_p$ is invertible. $\qquad\square$

**Lemma G.3.** $\mathcal{F} = \operatorname{span}\{\mathcal{K}_p(\cdot, x) \mid x \in \mathcal{X}\}$ *for $p$ non-vanishing on $\mathcal{V}$.*

*Proof.* From lemma G.2, $\operatorname{Im}\mathcal{K}_p = \mathcal{F}$. From the definition of the operator, $\operatorname{Im}\mathcal{K}_p \subset \operatorname{span}\{\mathcal{K}_p(\cdot, x) \mid x \in \mathcal{X}\}$. Then, the lemma follows. $\qquad\square$

**Lemma G.4.** *For non-vanishing distribution $p$ on $\mathcal{V}$*

$$\langle \mathcal{K}_p f, g \rangle_{\mathcal{H}(\mathcal{K}_p)} = \langle f, g \rangle_{L_2(\rho)}.$$

*Proof.* It is sufficient to check on the basis. So, we take $g = \mathcal{K}_p(\cdot, x)$. Then,

$$\langle \mathcal{K}_p f, \mathcal{K}_p(\cdot, x) \rangle_{\mathcal{H}(\mathcal{K}_p)} = \mathcal{K}_p[f](x) = \langle f, \mathcal{K}_p(\cdot, x) \rangle_{L_2(\rho)},$$

where the second equality holds by the definition of the operator $\mathcal{K}_p$. $\qquad\square$

For a weak learner $\nu$, we define a covariance operator:

$$\Sigma_\nu[f] = \frac{1}{N} k_\nu(\cdot, \mathbf{x}_N) f(\mathbf{x}_N), \ \Sigma_\nu : \mathcal{H} \to \mathcal{H}.$$

Also, for an arbitrary probability distribution $p$ over $\mathcal{V}$, we denote $\Sigma_p = \sum_{\nu \in \mathcal{V}} \Sigma_\nu p(\nu)$. These operators are typically referred to as covariance operators.

Let us formulate and prove several lemmas about the RKHS structure and operators $\Sigma, \Sigma_f, \Sigma_\nu$.

**Lemma G.5.** *(Courant-Fischer) Let $A$ be an $n \times n$ real symmetric matrix and $\lambda_1 \le \ldots \le \lambda_n$ its eigenvalues. Then,*

$$\lambda_k = \min_{dimU=k} \max_{x \in U} R_A(x),$$
$$\lambda_k = \max_{dimU=n-k+1} \min_{x \in U} R_A(x),$$

*where $R_A(x) = \frac{\langle Ax, x \rangle}{\langle x, x \rangle}$ is the Rayleigh-Ritz quotient.*

**Lemma G.6.** $\rho(k_\nu(\mathbf{x}_N, \mathbf{x}_N)) = \|k_\nu(\mathbf{x}_N, \mathbf{x}_N)\| = N$ *for any $\nu \in \mathcal{V}$.*

*Proof.* It is easy to see that

$$k_\nu(\mathbf{x}_N, \mathbf{x}_N) = \oplus_{i=1}^{L_\nu} w_\nu^{(i)} \mathbb{1}_{N_\nu^i \times N_\nu^i},$$

where $\mathbb{1}_{n \times n}$ is a matrix of size $n \times n$ consisting of ones. Then, we note that $\|\mathbb{1}_{n \times n}\| = n$ and now the statement of the lemma follows. $\qquad\square$

**Lemma G.7.** *(Covariation majorization) The following operator inequality holds for probability distributions $p, p'$ over $\mathcal{V}$, where $p$ is arbitrary and $p'$ does not vanish at any $\nu \in \mathcal{V}$:*

$$\lambda_{\max}(\Sigma_p) \le 1,$$
$$\lambda_{\min}(\Sigma_{p'}) \ge \frac{1}{N}.$$

*Proof.* Consider the following operators:

$$A : \mathcal{F} \to \mathbb{R}^n, \ f \mapsto f(\mathbf{x}_N),$$
$$B : \mathbb{R}^n \to \mathcal{F}, \ v \mapsto \mathcal{K}_p(\cdot, \mathbf{x}_N)v.$$

Then, $\Sigma_p = \frac{1}{N} BA$ and $K_N = \frac{1}{N}\mathcal{K}_p(\mathbf{x}_N, \mathbf{x}_N) = \frac{1}{N}AB$. As $AB$ and $BA$ have the same spectra, we further study the spectrum of $K_N$.

We have $K_N = \frac{1}{N}\mathcal{K}_p(\mathbf{x}_N, \mathbf{x}_N) = \sum_{\nu \in \mathcal{V}} \frac{1}{N} k_\nu(\mathbf{x}_N, \mathbf{x}_N)p(\nu)$ and $\lambda_{max}(K_N) \le 1$ follows from lemma G.6. Then, $\lambda_{max}(\Sigma_p) \le 1$ follows.

Now we need to show that $\lambda_{\min}(\Sigma_{p'}) \geq \frac{1}{N}$. Consider the following formula:

$$\Sigma_{p'} = \frac{1}{N} \sum_{i=1}^{N} \mathcal{K}_{p'}(\cdot, x_i) \otimes_{\mathcal{H}(\mathcal{K}_{p'})} \mathcal{K}_{p'}(\cdot, x_i),$$

$$\Sigma_{p'} = \frac{1}{N} \sum_{i=1}^{N} \mathcal{K}_{p'}(x_i, x_i) \Big( \frac{\mathcal{K}_{p'}(\cdot, x_i)}{\sqrt{\mathcal{K}_{p'}(x_i, x_i)}} \otimes_{\mathcal{H}(\mathcal{K}_{p'})} \frac{\mathcal{K}_{p'}(\cdot, x_i)}{\sqrt{\mathcal{K}_{p'}(x_i, x_i)}} \Big),$$

where $(a \otimes_{\mathcal{H}(\mathcal{K}_{p'})} b)[c] = \langle b, c \rangle_{\mathcal{H}(\mathcal{K}_{p'})} a$. If $a = b$ and $\|a\|_{\mathcal{H}(\mathcal{K}_{p'})} = 1$, then 1 and 0 are the only eigenvalues of $a \otimes_{\mathcal{H}(\mathcal{K}_{p'})} a$.

Denote by $S = span\{\mathcal{K}_{p'}(\cdot, x_i) \mid i = 1, \ldots, N\} \subset \mathcal{H}(\mathcal{K}_{p'})$ and $m = dimS$, $n = dim\mathcal{H}(\mathcal{K}_{p'})$. Then,

$$\lambda_{min}(\Sigma_{p'}) = \lambda_{n-m+1}(\Sigma_{p'}) = \min_{dimU = n-m+1} \max_{x \in U} R_{\Sigma_{p'}}(x),$$

where $R_{\Sigma_{p'}}(x) = \frac{(\Sigma_{p'}x, x)_{\mathcal{H}(\mathcal{K}_{p'})}}{(x, x)_{\mathcal{H}(\mathcal{K}_{p'})}}$. As $dimU = n - m + 1$, then $U \cap S \neq \varnothing$. Suppose $\mathcal{K}_{p'}(\cdot, x_i) \in U \cap S$, then

$$\max_{x \in U} R_{\Sigma_{p'}}(x) \geq R_{\Sigma_{p'}} \Big( \frac{\mathcal{K}_{p'}(\cdot, x_i)}{\sqrt{\mathcal{K}_{p'}(x_i, x_i)}} \Big) \overset{*}{\geq} \frac{\mathcal{K}_{p'}(x_i, x_i)}{N} \geq \frac{1}{N},$$

where (*) is fulfilled as $a \otimes_{\mathcal{H}(\mathcal{K}_{p'})} a$ is a positive semidefinite operator and the last inequality follows from $\mathcal{K}_{p'}(x, x) \geq 1 \, \forall x \in \mathcal{X}$.

$\square$

## G.2 NORM MAJORIZATION

The following lemmas relate the norms $L_2, \mathcal{H}, \mathbb{R}^N$ with respect to each other.[9] Indeed, by these lemmas we can consider the bound $\| \cdot \|_{L_2} \lesssim \| \cdot \|_{\mathcal{H}} \leq \| \cdot \|_{\mathbb{R}^N}$. Further, in the main theorems we will use these relations extensively.

**Corollary G.8.** $\|f(\mathbf{x}_N)\| \geq \|f\|_{\mathcal{H}}$ for $f \in span\{\mathcal{K}(\cdot, x_i) \mid i = 1, \ldots, N\}$.

*Proof.*

$$\frac{1}{N} \|f(\mathbf{x}_N)\|^2 = \langle \Sigma f, f \rangle_{\mathcal{H}} \geq \frac{1}{N} \|f\|_{\mathcal{H}}^2$$

as $\Sigma \succeq \frac{1}{N} I$ on $span\{\mathcal{K}(\cdot, x_i) \mid i = 1, \ldots, N\}$.  $\square$

**Lemma G.9.** $\lambda_{max}(\mathcal{K}) \leq \max_{x \in \mathcal{X}} \mathcal{K}(x, x)$.

*Proof.* Consider $\mathcal{K}$ as an operator on $(\mathcal{F}, L_2(\rho))$. We will prove that $\|\mathcal{K}\|_{\mathcal{B}((\mathcal{F}, L_2(\rho)))} \leq \max_{x \in \mathcal{X}} \mathcal{K}(x, x)$ and the lemma will follow. Consider the inequality

$$\mathcal{K}[f](x) = \langle \mathcal{K}_x, f \rangle_{L_2} \leq \|\mathcal{K}_x\|_{L_2} \|f\|_{L_2}.$$

Then, $\|\mathcal{K}[f]\|_{L_2} \leq \max_{x \in \mathcal{X}} \|\mathcal{K}_x\|_{L_2} \|f\|_{L_2}$. Note also that $\mathcal{K}(x, x') \leq \min(\mathcal{K}(x, x), \mathcal{K}(x', x'))$ which can be easily seen from the definition. Then, $\|\mathcal{K}_x\|_{L_2} \leq \mathcal{K}(x, x)$ and the lemma follows.  $\square$

**Corollary G.10.** *(Expected squared norm majorization by RKHS norm) The following bound holds $\forall f \in \mathcal{H}$:*

$$\|f\|_{L_2(\rho)}^2 \leq \max_{x \in \mathcal{X}} \mathcal{K}(x, x) \|f\|_{\mathcal{H}}^2.$$

*Proof.* We have

$$\lambda_{max}(\mathcal{K}) \|f\|_{\mathcal{H}}^2 \geq \langle \mathcal{K}f, f \rangle_{\mathcal{H}} = \langle f, f \rangle_{L_2} = \|f\|_{L_2}^2.$$

Then, from the previous lemma the bound holds.  $\square$

---

[9]Note that $\| \cdot \|_{\mathbb{R}^N}$ indeed becomes a norm once we restrict our space to $span\{\mathcal{K}(\cdot, x_i) \mid i = 1, \ldots, N\}$.

### G.3 SYMMETRY OF OPERATORS

In this section, we establish symmetry of various operators with respect to the norms $L_2, \mathcal{H}, \mathbb{R}^N$. These results are mainly required to claim that the spectral radii of these operators coincide with their operator norms in various spaces: $B(\mathcal{F}, L_2), B(\mathcal{H}), B(span\{\mathcal{K}(\cdot, x_i) \mid i = 1, \ldots, N\}, \|\cdot\|_{\mathbb{R}^N})$. Though, we use symmetry of operators not only this way.

**Lemma G.11.** *(Universal symmetry of covariance operators in $\mathcal{H}$) The operator $\Sigma_p$ for any $p$ is symmetric w.r.t. the dot product of $\mathcal{H}(\mathcal{K}_{p'})$ for any non-singular $p'$.*

*Proof.* First, let us prove the statement for non-singular distribution $p$. To see that, we consider the following quantity:

$$\langle \Sigma_p f, g \rangle_{\mathcal{H}(\mathcal{K}_p)} = \frac{1}{N} \sum_{i=1}^{N} \langle \mathcal{K}_p(\cdot, x_i), f \rangle_{\mathcal{H}(\mathcal{K}_p)} \langle \mathcal{K}_p(\cdot, x_i), g \rangle_{\mathcal{H}(\mathcal{K}_p)}.$$

Then, we use the following trick: $\langle \mathcal{K}_p(\cdot, x_i), f \rangle_{\mathcal{H}(\mathcal{K}_p)} = \langle \mathcal{K}_{p'} \mathcal{K}_p^{-1} \mathcal{K}_p(\cdot, x_i), f \rangle_{\mathcal{H}(\mathcal{K}_{p'})}$. It allows us to rewrite:

$$\langle \Sigma_p f, g \rangle_{\mathcal{H}(\mathcal{K}_p)} = \frac{1}{N} \sum_{i=1}^{N} \langle \mathcal{K}_{p'} \mathcal{K}_p^{-1} \mathcal{K}_p(\cdot, x_i), f \rangle_{\mathcal{H}(\mathcal{K}_{p'})} \langle \mathcal{K}_{p'} \mathcal{K}_p^{-1} \mathcal{K}_p(\cdot, x_i), g \rangle_{\mathcal{H}(\mathcal{K}_{p'})}.$$

From this, it immediately follows:

$$\Sigma_p = \frac{1}{N} \sum_{i=1}^{N} \left( \mathcal{K}_{p'} \mathcal{K}_p^{-1} \mathcal{K}_p(\cdot, x_i) \right) \otimes_{\mathcal{H}(\mathcal{K}_{p'})} \left( \mathcal{K}_{p'} \mathcal{K}_p^{-1} \mathcal{K}_p(\cdot, x_i) \right).$$

This shows that $\Sigma_p$ is indeed symmetric w.r.t. the dot product of $\mathcal{H}(\mathcal{K}_{p'})$. Finally, we can use the continuity argument, which we can use due to intrinsic finite dimension, to conclude that symmetry must hold for arbitrary distributions, in particular for $p = \delta_\nu$ which corresponds to $\Sigma_\nu$. $\qquad\square$

**Lemma G.12.** *(Universal symmetry of covariance operators in $L_2$) The operator $\Sigma_p$ for any $p$ is symmetric w.r.t. the dot product of $L_2$.*

*Proof.* We consider similarly the following quantity:

$$\langle \Sigma_p f, g \rangle_{\mathcal{H}(\mathcal{K}_p)} = \frac{1}{N} \sum_{i=1}^{N} \langle \mathcal{K}_p(\cdot, x_i), f \rangle_{\mathcal{H}(\mathcal{K}_p)} \langle \mathcal{K}_p(\cdot, x_i), g \rangle_{\mathcal{H}(\mathcal{K}_p)}.$$

Then, we use the following trick $\langle \mathcal{K}_p(\cdot, x_i), f \rangle_{\mathcal{H}(\mathcal{K}_p)} = \langle \mathcal{K}_p^{-1} \mathcal{K}_p(\cdot, x_i), f \rangle_{L_2}$. It allows us to rewrite:

$$\langle \Sigma_p f, g \rangle_{\mathcal{H}(\mathcal{K}_p)} = \frac{1}{N} \sum_{i=1}^{N} \langle \mathcal{K}_p^{-1} \mathcal{K}_p(\cdot, x_i), f \rangle_{L_2} \langle \mathcal{K}_p^{-1} \mathcal{K}_p(\cdot, x_i), g \rangle_{L_2}.$$

From this, it immediately follows:

$$\Sigma_p = \frac{1}{N} \sum_{i=1}^{N} \left( \mathcal{K}_p^{-1} \mathcal{K}_p(\cdot, x_i) \right) \otimes_{L_2} \left( \mathcal{K}_p^{-1} \mathcal{K}_p(\cdot, x_i) \right)$$

Which shows that $\Sigma_p$ is indeed symmetric w.r.t. the dot product of $L_2$. $\qquad\square$

**Lemma G.13.** *(Universal symmetry of kernel operators in $\mathcal{H}$) The operator $\mathcal{K}_p$ for any $p$ is symmetric w.r.t. the dot product of $\mathcal{H}$.*

*Proof.* We consider decomposing $\mathcal{K}_p$ as:

$$\langle \mathcal{K}_p f, g \rangle_{L_2} = \int_{\mathcal{X} \times \mathcal{X}} \mathcal{K}_p(x, y) f(x) g(y) \rho(\mathrm{d}x) \rho(\mathrm{d}y).$$

Then, we use the following trick: $f(x) = \langle \mathcal{K}(\cdot, x), f \rangle_{\mathcal{H}}$. It allows us to rewrite:

$$\langle \mathcal{K}_p f, g \rangle_{L_2} = \int_{\mathcal{X} \times \mathcal{X}} \mathcal{K}_p(x, y) \langle \mathcal{K}(\cdot, x), f \rangle_{\mathcal{H}} \langle \mathcal{K}(\cdot, y), g \rangle_{\mathcal{H}} \rho(dx)\rho(dy).$$

From this, it immediately follows:

$$\mathcal{K}_p = \int_{\mathcal{X} \times \mathcal{X}} \mathcal{K}_p(x, y)\Big(\mathcal{K}(\cdot, x) \otimes_{\mathcal{H}} \mathcal{K}(\cdot, y)\Big)\rho(dx)\rho(dy).$$

This shows that $\mathcal{K}_p$ is indeed symmetric w.r.t. the dot product of $\mathcal{H}$ since both $\mathcal{K}(\cdot, x) \otimes_{\mathcal{H}} \mathcal{K}(\cdot, y)$ and $\mathcal{K}(\cdot, y) \otimes_{\mathcal{H}} \mathcal{K}(\cdot, x)$ are present with the same weight $\mathcal{K}_p(x, y)\rho(dx)\rho(dy) = \mathcal{K}_p(y, x)\rho(dy)\rho(dx)$. $\square$

### G.4 ITERATIONS OF GRADIENT BOOSTING

**Lemma G.14.** *For any $\nu \in \mathcal{V}$, we have $k_\nu(\cdot, \mathbf{x}_N)[\mathbf{y}_N - f_*(\mathbf{x}_N)] = 0$.*

*Proof.* Follows from Lemma D.5. $\square$

**Lemma G.15** (Lemma 3.7 in the main text)**.** *Iterations $f_\tau$ of gradient boosting (Algorithm 2) can be written in the form:*

$$f_{\tau+1} = \left(1 - \frac{\lambda\epsilon}{N}\right)f_\tau + \frac{\epsilon}{N}k_{\nu_\tau}(\cdot, \mathbf{x}_N)\big[\mathbf{y}_N - f_\tau(\mathbf{x}_N)\big] = (1 - \frac{\lambda\epsilon}{N})f_\tau + \frac{\epsilon}{N}k_{\nu_\tau}(\cdot, \mathbf{x}_N)\big[f_*(\mathbf{x}_N) - f_\tau(\mathbf{x}_N)\big],$$

$$\nu_\tau \sim p(\nu|f_\tau, \beta).$$

*Proof.* According to Algorithm 2:

$$f_{\tau+1}(\cdot) = \big(1 - \frac{\lambda\epsilon}{N}\big)f_\tau(\cdot) + \epsilon\big\langle \phi_{\nu_\tau}(\cdot), \theta_\tau \big\rangle_{\mathbb{R}^{L_{\nu_\tau}}} \text{ for } \theta_\tau = \Big(\frac{\sum_{i=1}^N \phi_{\nu_\tau}^{(j)}(x_i)r_\tau^{(i)}}{\sum_{i=1}^N \phi_{\nu_\tau}^{(j)}(x_i)}\Big)_{j=1}^{L_{\nu_\tau}}.$$

Thus,

$$f_{\tau+1} = \big(1 - \frac{\lambda\epsilon}{N}\big)f_\tau + \epsilon\frac{1}{N}\sum_{j=1}^{L_{\nu_\tau}}\omega_{\nu_\tau}^j\phi_{\nu_\tau}^j\sum_{i:\ \phi_{\nu_\tau}^j(x_i)=1}r_\tau^i.$$

Now note that $k_{\nu_\tau}(\cdot, x_i) = \omega_{\nu_\tau}^j\phi_{\nu_\tau}^j(\cdot)$, where $j$ is such that $\phi_{\nu_\tau}^j(x_i) = 1$. From this the lemma follows. $\square$

From Lemmas G.15, D.4, it is easy to show that $f_\tau, f_* \in span\{\mathcal{K}(\cdot, x_i) \mid i = 1, \ldots, N\}$. Then, hereafter we can use Corollary G.8 to bound $\mathcal{H}$ norm with $\mathbb{R}^N$ norm.

**Lemma G.16.** *The iterations of gradient boosting can be represented as:*

$$\epsilon^{-1}\mathbb{E}\big(f_\tau - f_{\tau+1}\big) \mid f_\tau = \mathcal{K}_{f_\tau}D[f_\tau - f_*] + \frac{\lambda}{N}f_\tau,$$

*where $D : \mathcal{F} \to \mathcal{F}$ is bounded linear operator defined as Riesz representative with respect to $L_2$ scalar product of such bilinear function $\frac{1}{N}f(\mathbf{x}_N)^T h(\mathbf{x}_N) = \langle Df, h \rangle_{L_2}$. Similar decomposition holds for $\nabla_{\mathcal{H}}V(f, \lambda) = \mathcal{K}D[f - f_*] + \frac{\lambda}{N}f$.*

*Proof.* First, observe that $\mathbb{E}f_{\tau+1} \mid f_\tau = f_\tau - \epsilon\nabla V(f_\tau, \lambda)$ where gradient here is taken with respect to $\mathcal{H}(\mathcal{K}_{f_\tau})$. Keep in mind that in the definition of $V(f_\tau, \lambda)$, the norm in the regularizer term is taken with respect to $\mathcal{H}(\mathcal{K}_{f_\tau})$ instead of $\mathcal{H}(\mathcal{K})$. Thus, we need only to prove that $\nabla_{\mathcal{H}}V(f, \lambda) = \mathcal{K}D[f - f_*] + \frac{\lambda}{N}f$.

Consider Fréchet differential $\mathcal{D}_f V(f)[h] = \frac{1}{N}(f(\mathbf{x}_N) - f_*(\mathbf{x}_N))^T h(\mathbf{x}_N) = \langle D[f - f_*], h \rangle_{L_2}$. By Lemma G.4, we deduce

$$\mathcal{D}_f V(f)[h] = \frac{1}{N}(f(\mathbf{x}_N) - f_*(\mathbf{x}_N))^T h(\mathbf{x}_N) = \langle \mathcal{K}D[f - f_*], h \rangle_{\mathcal{H}},$$

which implies $\nabla V(f) = \mathcal{K}D[f - f_*]$ and the lemma follows.

$\square$

### G.5 MAIN LEMMAS

**Lemma G.17.** *Let $A, B \in \mathcal{B}(\mathcal{H}, \mathcal{H})$ be two PSD operators such that $\xi B - A$ and $\xi A - B$ are PSD for some $\xi \in (1, \infty)$. Let $g, h \in \mathcal{H}$ be two arbitrary vectors and $\lambda \in \mathbb{R}_{++}$ be a constant. Then,*

$$\langle A[g] + \lambda \xi h, B[g] + \lambda h \rangle_{\mathcal{H}} \geq \frac{1}{2}\Big(\xi^{-1}\big\|A[g] + \lambda \xi h\big\|_{\mathcal{H}}^2 - \xi(\xi^2 - 1)\lambda^2 \|h\|_{\mathcal{H}}^2\Big).$$

*Proof.* Consider the following equality:

$$\xi\langle\xi^{-1}A[g] + \lambda h, B[g] + \lambda h\rangle_{\mathcal{H}} = \frac{\xi}{2}\Big(\big\|\xi^{-1}A[g] + \lambda h\big\|_{\mathcal{H}}^2 + \big\|B[g] + \lambda h\big\|_{\mathcal{H}}^2 - \big\|(B - \xi^{-1}A)[g]\big\|_{\mathcal{H}}^2\Big),$$

which is basically the classical decomposition of the dot product $\langle x, y \rangle = \frac{1}{2}\big(\|x\|^2 + \|y\|^2 - \|x - y\|^2\big)$. Then, we note that $(1 - \xi^{-2})B - (B - \xi^{-1}A) = \xi^{-2}(\xi A - B)$ is PSD by assumption and since $(B - \xi^{-1}A)$ is PSD it implies that $(1 - \xi^{-2})B \geq (B - \xi^{-1}A)$, which implies:

$$\langle A[g] + \lambda \xi h, B[g] + \lambda h\rangle_{\mathcal{H}} \geq \frac{\xi}{2}\Big(\big\|\xi^{-1}A[g] + \lambda h\big\|_{\mathcal{H}}^2 + \big\|B[g] + \lambda h\big\|_{\mathcal{H}}^2 - (1 - \xi^{-2})^2\big\|B[g]\big\|_{\mathcal{H}}^2\Big).$$

Finally, note that $\frac{\xi^2}{2 - \xi^{-2}} - 1 = \frac{\xi^2(1 - \xi^{-2})^2}{2 - \xi^{-2}} \leq \xi^2 - 1$. Then, the result directly follows from the following equality:

$$\big\|B[g] + \lambda h\big\|_{\mathcal{H}}^2 - (1 - \xi^{-2})^2\big\|B[g]\big\|_{\mathcal{H}}^2$$
$$= \big\|\xi^{-1}\sqrt{2 - \xi^{-2}}B[g] + \frac{\lambda}{\xi^{-1}\sqrt{2 - \xi^{-2}}}h\big\|_{\mathcal{H}}^2 - \lambda^2\big(\frac{\xi^2}{2 - \xi^{-2}} - 1\big)\|h\|_{\mathcal{H}}^2.$$

$\square$

Denote $\kappa(A, B) = \|\mathrm{Id}_{\mathcal{H}} - BA^{-1}\|_{\mathcal{B}(\mathcal{H},\mathcal{H})} = \|(B - A)A^{-1}\|_{\mathcal{B}(\mathcal{H},\mathcal{H})}$.

**Lemma G.18.** *If $\xi^{-1}\mathcal{K} \preceq \mathcal{K}_f \preceq \xi\mathcal{K}$ for $\xi > 1$, then $\kappa(\mathcal{K}, \mathcal{K}_f) \leq \xi - 1$.*

*Proof.* First, we note that both operators are symmetric semi-positive definite in $L_2$. Now, let us look at the Rayleigh quotient:

$$\|(\mathcal{K} - \mathcal{K}_f)\mathcal{K}^{-1}\|_{\mathcal{B}(\mathcal{H},\mathcal{H})} = \max_{f \in \mathcal{F} \backslash \{\mathbf{0}\}} \frac{\|(\mathcal{K} - \mathcal{K}_f)\mathcal{K}^{-1}f\|_{\mathcal{H}}}{\|f\|_{\mathcal{H}}} = \max_{f \in \mathcal{F} \backslash \{\mathbf{0}\}} \frac{\|\mathcal{K}^{-\frac{1}{2}}(\mathcal{K} - \mathcal{K}_f)\mathcal{K}^{-\frac{1}{2}}f\|_{L_2}}{\|f\|_{L_2}}.$$

In the last equality we used fact that $\mathcal{K}$ is symmetric positive definite and therefore $\mathcal{K}^{\frac{1}{2}}$ is too and hence we can substitute $f \leftarrow \mathcal{K}^{\frac{1}{2}}f$ and use the explicit formula for the dot product in $\mathcal{H}$ via the product in $L_2$. Now we observe that

$$-(\xi - 1)\mathrm{Id}_{L_2} = \mathcal{K}^{-\frac{1}{2}}(\mathcal{K} - \xi\mathcal{K})\mathcal{K}^{-\frac{1}{2}} \preceq \mathcal{K}^{-\frac{1}{2}}(\mathcal{K} - \mathcal{K}_f)\mathcal{K}^{-\frac{1}{2}}$$
$$\preceq \mathcal{K}^{-\frac{1}{2}}(\mathcal{K} - \xi^{-1}\mathcal{K})\mathcal{K}^{-\frac{1}{2}} = (1 - \xi^{-1})\mathrm{Id}_{L_2} \preceq (\xi - 1)\mathrm{Id}_{L_2},$$

which implies that the spectral radius $\rho(\mathcal{K}^{-\frac{1}{2}}(\mathcal{K} - \mathcal{K}_f)\mathcal{K}^{-\frac{1}{2}})$ is bounded by $\xi - 1$. Therefore, we obtain:

$$\max_{f \in \mathcal{F} \backslash \{\mathbf{0}\}} \frac{\|\mathcal{K}^{-\frac{1}{2}}(\mathcal{K} - \mathcal{K}_f)\mathcal{K}^{-\frac{1}{2}}f\|_{L_2}}{\|f\|_{L_2}} = \rho(\mathcal{K}^{-\frac{1}{2}}(\mathcal{K} - \mathcal{K}_f)\mathcal{K}^{-\frac{1}{2}}) \leq \xi - 1.$$

$\square$

**Lemma G.19.** *Let $A, B \in \mathcal{B}(\mathcal{H}, \mathcal{H})$. Then, the following inequality holds:*

$$\langle A[g] + \lambda h, B[g] + \lambda h\rangle_{\mathcal{H}} \geq \big(\frac{1}{2} - \kappa(A, B)\big)\big\|A[g] + \lambda h\big\|_{\mathcal{H}}^2 - \kappa^2(A, B)\frac{\lambda^2}{2}\|h\|_{\mathcal{H}}^2.$$

*Proof.* Let us rewrite the left part as

$$\langle A[g]+\lambda h, B[g]+\lambda h\rangle_{\mathcal{H}} = \langle A[g]+\lambda h, \left(\mathrm{Id}_{L_2}+(BA^{-1}-\mathrm{Id}_{L_2})\right)(A[g]+\lambda h)+\lambda(\mathrm{Id}_{L_2}-BA^{-1})h\rangle_{\mathcal{H}}$$

$$= \|A[g]+\lambda h\|_{\mathcal{H}}^2 - \langle A[g]+\lambda h, (\mathrm{Id}_{L_2}-BA^{-1})(A[g]+\lambda h)\rangle_{\mathcal{H}} + \langle A[g]+\lambda h, \lambda(\mathrm{Id}_{L_2}-BA^{-1})h\rangle_{\mathcal{H}}.$$

Then, we use the equality $\langle a,b\rangle = \frac{1}{2}\left(\|a\|^2+\|b\|^2-\|a-b\|^2\right)$. Also, we use

$$\langle A[g]+\lambda h, (Id_{L_2}-BA^{-1})(A[g]+\lambda h)\rangle_{\mathcal{H}}$$
$$\leq \|A[g]+\lambda h\|_{\mathcal{H}}\|(Id_{L_2}-BA^{-1})(A[g]+\lambda h)\|_{\mathcal{H}}$$
$$\leq \kappa(A,B)\|A[g]+\lambda h\|_{\mathcal{H}}^2$$

to obtain

$$\langle A[g]+\lambda h, B[g]+\lambda h\rangle_{\mathcal{H}} \geq \left(\frac{3}{2}-\kappa(A,B)\right)\|A[g]+\lambda h\|_{\mathcal{H}}^2$$

$$+ \frac{\lambda^2}{2}\|(\mathrm{Id}_{L_2}-BA^{-1})h\|_{\mathcal{H}}^2 - \frac{1}{2}\|(A[g]+\lambda h)-\lambda(\mathrm{Id}_{L_2}-BA^{-1})h\|_{\mathcal{H}}^2$$

$$\geq \left(\frac{3}{2}-\kappa(A,B)\right)\|A[g]+\lambda h\|_{\mathcal{H}}^2 - \|(A[g]+\lambda h)\|_{\mathcal{H}}^2 - \frac{\lambda^2}{2}\|(\mathrm{Id}_{L_2}-BA^{-1})h\|_{\mathcal{H}}^2$$

$$\geq \left(\frac{1}{2}-\kappa(A,B)\right)\|A[g]+\lambda h\|_{\mathcal{H}}^2 - \kappa^2(A,B)\frac{\lambda^2}{2}\|h\|_{\mathcal{H}}^2.$$

$\square$

The following lemma holds.

**Lemma G.20.** *If $(\frac{\lambda}{N}+1)\epsilon < 1$ and $f_0 = \mathbb{0}_{\mathcal{H}}$, then $\forall\tau$ the following holds almost surely*

$$\|f_\tau - f_*\| \leq \|f_*\|$$

*for norms $\|\cdot\|_{L_2}$, $\|\cdot\|_{\mathcal{H}}$, and $\|\cdot\|_{\mathbb{R}^N}$.*

*Proof.* Note that

$$f_{\tau+1} - f_* = \left(\mathrm{Id}_{L_2} - \frac{\lambda\epsilon}{N}\mathrm{Id}_{L_2} - \epsilon\Sigma_{\nu_\tau}\right)[f_\tau - f_*] - \frac{\lambda\epsilon}{N}f_*.$$

Now observe that $S := \left(\mathrm{Id}_{L_2} - \frac{\lambda\epsilon}{N}\mathrm{Id}_{L_2} - \epsilon\Sigma_{\nu_\tau}\right)$ is symmetric with eigenvalues $0 < \lambda_i(S) \leq 1 - \frac{\lambda\epsilon}{N}$, therefore its operator norm in $\mathcal{B}(L_2)$, $\mathcal{B}(\mathcal{H})$, and $\mathbb{R}^{N\times N}$ is less than $1 - \frac{\lambda\epsilon}{N}$. Taking the norm of left and right sides and using the sub-additivity of the norm, we obtain:

$$\|f_{\tau+1} - f_*\| \leq (1-\frac{\lambda\epsilon}{N})\|f_\tau - f_*\| + \frac{\lambda\epsilon}{N}\|f_*\|.$$

Since $\|f_0 - f_*\| = \|f_*\|$ that recurrent relation inductively yields the statement of the lemma. $\square$

**Corollary G.21.** *Under the same conditions, $\|f_\tau\| \leq 2\|f_*\|$.*

## G.6 MAIN THEOREMS

Let us denote $R = \|f_*\|_{\mathbb{R}^N}$. We argue that it is a constant value since the kernel $\mathcal{H}$ and $f_*$ are convergent as $N \to \infty$ which makes it bounded by some constant with probability arbitrary close to one. By Lemma F.2, $\Gamma_\beta(f) \leq e^{\frac{2mV(f)}{\beta}}$. Then, $\Gamma_\beta(f_\tau) \leq e^{\frac{2m\frac{1}{2N}\|f_\tau - f_*\|_{\mathbb{R}^N}^2}{\beta}} \leq e^{\frac{mR^2}{N\beta}}$ and we denote $M_\beta = e^{\frac{mR^2}{N\beta}} > 1$.

**Theorem G.22.** *Consider an arbitrary $\epsilon$, $0 < \epsilon(\frac{\lambda}{N}+1) < 1$ and $\frac{(1+M_\beta\lambda)}{4M_\beta N} \geq \epsilon$. The following inequality holds:*

$$\mathbb{E}V(f_T) \leq \frac{R^2}{2N}e^{-\frac{1+M_\beta\lambda}{2M_\beta N}T\epsilon}$$

$$+ M_\beta\lambda\left(\frac{1}{2N} + \frac{4M_\beta}{1+M_\beta\lambda}(M_\beta^2-1)\frac{\lambda}{N} + \frac{4\epsilon}{1+M_\beta\lambda}\left(\frac{2\lambda}{N^2} + M_\beta(1+\frac{2\lambda^2}{N^2})\right)\right)R^2.$$

*Proof.* To prove the theorem, we will bound $V(f) \leq V(f, M_\beta \lambda) + const.$ It will allow us to invoke Lemma G.17. After that by using strong convexity we obtain a bound on $\mathbb{E}V(f_\tau, M_\beta)$ and then a bound on $\mathbb{E}V(f_\tau)$ will follow straightforwardly.

To get the result for $V(f_\tau, M_\beta \lambda)$, we expand $V(f_{\tau+1}, M_\beta \lambda)$ by substituting the formula for $f_{\tau+1}$ and first dealing with the term $V(f_{\tau+1})$:

$$\frac{1}{2N}\|f_{\tau+1}(\mathbf{x}_N) - f_*(\mathbf{x}_N)\|_{\mathbb{R}^N}^2 = \frac{1}{2N}\|f_\tau(\mathbf{x}_N) - f_*(\mathbf{x}_N)\|_{\mathbb{R}^N}^2 + \frac{1}{N}\langle f_{\tau+1}(\mathbf{x}_N)$$
$$- f_\tau(\mathbf{x}_N), f_\tau(\mathbf{x}_N) - f_*(\mathbf{x}_N)\rangle_{\mathbb{R}^N} + \frac{1}{2N}\|f_{\tau+1}(\mathbf{x}_N) - f_\tau(\mathbf{x}_N)\|_{\mathbb{R}^N}^2$$
$$= V(f_\tau) - \epsilon\langle\Sigma_{\nu_\tau}[f_\tau - f_*] + \frac{\lambda}{N}f_\tau, \mathcal{K}D[f_\tau - f_*]\rangle_{\mathcal{H}} + \frac{\epsilon^2}{2N}\|\Sigma_{\nu_\tau}[f_\tau - f_*] + \frac{\lambda}{N}f_\tau\|_{\mathbb{R}^N}^2$$
$$\leq V(f_\tau) - \epsilon\langle\Sigma_{\nu_\tau}[f_\tau - f_*] + \frac{\lambda}{N}f_\tau, \mathcal{K}D[f_\tau - f_*]\rangle_{\mathcal{H}} + 2\epsilon^2 V(f_\tau) + \frac{\lambda^2\epsilon^2}{N^3}\|f_\tau(\mathbf{x}_N)\|_{\mathbb{R}^N}^2,$$

where we used the inequality $\|a+b\|^2 \leq 2\|a\|^2 + 2\|b\|^2$ and Lemma G.7 which allows us to bound $\|\Sigma_{\nu_\tau}(f_\tau - f_*)\|_{\mathbb{R}^N} \leq \|(f_\tau - f_*)\|_{\mathbb{R}^N}$. Then, we analyze the regularization:

$$\frac{\lambda}{2N}M_\beta\|f_{\tau+1}\|_{\mathcal{H}}^2 = \frac{\lambda}{2N}M_\beta\|f_\tau\|_{\mathcal{H}}^2 + \langle f_{\tau+1} - f_\tau, \frac{\lambda}{N}M_\beta f_\tau\rangle_{\mathcal{H}} + \frac{\lambda\epsilon^2}{2N}M_\beta\|\Sigma_{\nu_\tau}[f_\tau - f_*] + \frac{\lambda}{N}f_\tau\|_{\mathcal{H}}^2$$
$$\leq \frac{\lambda}{2N}M_\beta\|f_\tau\|_{\mathcal{H}}^2 - \epsilon\langle\frac{\lambda}{N}M_\beta f_\tau, \Sigma_{\nu_\tau}[f_\tau - f_*] + \frac{\lambda}{N}f_\tau\rangle_{\mathcal{H}} + \frac{\epsilon^2\lambda}{N}M_\beta\|f_\tau - f_*\|_{\mathcal{H}}^2 + \frac{\epsilon^2\lambda^3}{N^3}M_\beta\|f_\tau\|_{\mathcal{H}}^2$$
$$\leq \frac{\lambda}{2N}M_\beta\|f_\tau\|_{\mathcal{H}}^2 - \epsilon\langle\frac{\lambda}{N}M_\beta f_\tau, \Sigma_{\nu_\tau}[f_\tau - f_*] + \frac{\lambda}{N}f_\tau\rangle_{\mathcal{H}} + \frac{\epsilon^2\lambda}{N}M_\beta(1 + \frac{4\lambda^2}{N^2})\|f_*\|_{\mathcal{H}}^2.$$

where in the first inequality we used $\|\Sigma_{\nu_\tau}[f_\tau - f_*]\|_{\mathcal{H}} \leq \|f_\tau - f_*\|_{\mathcal{H}}$ which is due to Lemma G.7. Summing up the expectations of those two expressions, we obtain:

$$\mathbb{E}V(f_{\tau+1}, M_\beta\lambda) = \mathbb{E}V(f_{\tau+1}) + \frac{M_\beta\lambda}{2N}\mathbb{E}\|f_{\tau+1}\|_{\mathcal{H}}^2 - C_{M_\beta\lambda}$$
$$\leq \mathbb{E}V(f_\tau) - \epsilon\mathbb{E}\langle\Sigma_{f_\tau}[f_\tau - f_*] + \frac{\lambda}{N}f_\tau, \mathcal{K}D[f_\tau - f_*] + \frac{M_\beta\lambda}{N}f_\tau\rangle_{\mathcal{H}} + 2\epsilon^2\mathbb{E}V(f_\tau)$$
$$+ \frac{\lambda^2\epsilon^2}{N^3}\mathbb{E}\|f_\tau(\mathbf{x}_N)\|_{\mathbb{R}^N}^2 + \frac{\lambda}{2N}M_\beta\mathbb{E}\|f_\tau\|_{\mathcal{H}}^2 + \frac{\epsilon^2\lambda}{N}M_\beta(1 + \frac{4\lambda^2}{N^2})\|f_*\|_{\mathcal{H}}^2 - C_{M_\beta\lambda}$$
$$\leq (1 + 2\epsilon^2)\left(\mathbb{E}V(f_\tau) + \frac{\lambda}{2N}M_\beta\mathbb{E}\|f_\tau\|_{\mathcal{H}}^2 - C_{M_\beta\lambda}\right) - \epsilon\mathbb{E}\langle\Sigma_{f_\tau}[f_\tau - f_*] + \frac{\lambda}{N}f_\tau, \mathcal{K}D[f_\tau - f_*]$$
$$+ \frac{M_\beta\lambda}{N}f_\tau\rangle_{\mathcal{H}} + \frac{\lambda^2\epsilon^2}{N^3}\mathbb{E}\|f_\tau(\mathbf{x}_N)\|_{\mathbb{R}^N}^2 + \frac{\epsilon^2\lambda}{N}M_\beta(1 + \frac{4\lambda^2}{N^2})\|f_*\|_{\mathcal{H}}^2 + 2\epsilon^2 C_{M_\beta\lambda}$$
$$\leq (1 + 2\epsilon^2)\mathbb{E}V(f_\tau, M_\beta\lambda) - \epsilon\mathbb{E}\langle\mathcal{K}_{f_\tau}D[f_\tau - f_*]$$
$$+ \frac{\lambda}{N}f_\tau, \mathcal{K}D[f_\tau - f_*] + \frac{M_\beta\lambda}{N}f_\tau\rangle + \frac{2\epsilon^2\lambda}{N}\left(\frac{2\lambda}{N^2} + M_\beta(1 + \frac{2\lambda^2}{N^2})\right)R^2.$$

Here we used

$$C_\lambda = \inf_{f \in F} L(f, \lambda) - \inf_{f \in F} L(f) \leq L(f_*, \lambda) - L(f_*) = \frac{\lambda}{2N}\|f_*\|_{\mathcal{H}}^2 \leq \frac{\lambda}{2N}R^2.$$

Then, by applying Lemma G.16 for $\Sigma_{f_\tau} = \mathcal{K}_{f_\tau}D$ and applying Lemma G.17 with $\xi = M_\beta$, $A = \mathcal{K}$, $B = \mathcal{K}_{f_\tau}$, $g = D[f_\tau - f_*]$, and $h = f_\tau$, we obtain the following bound:

$$-\epsilon\mathbb{E}\langle\mathcal{K}_{f_\tau}D[f_\tau - f_*] + \frac{\lambda}{N}f_\tau, \mathcal{K}D[f_\tau - f_*] + \frac{M_\beta\lambda}{N}f_\tau\rangle_{\mathcal{H}}$$
$$\leq -\frac{\epsilon}{2M_\beta}E\|\nabla_{\mathcal{H}}V(f_\tau, M_\beta\lambda)\|_{\mathcal{H}}^2 + \frac{\epsilon}{2}M_\beta(M_\beta^2 - 1)\frac{\lambda^2}{N^2}E\|f_\tau\|_{\mathcal{H}}^2$$
$$\leq -\frac{\epsilon}{2M_\beta}E\|\nabla_{\mathcal{H}}V(f_\tau, M_\beta\lambda)\|_{\mathcal{H}}^2 + 2\epsilon M_\beta(M_\beta^2 - 1)\frac{\lambda^2}{N^2}R^2.$$

Then, by using Polyak-Łojasiewicz inequality $\frac{1}{2}\|\nabla V\|_{\mathcal{H}} \geq \mu V$ for $\mu$-strongly convex function $V$ (restricted on $span\{\mathcal{K}(\cdot, x_i) \mid i = 1, \ldots, N\}$) with $\mu \geq \frac{1+M_\beta\lambda}{N} > 0$, which is due to Corollary G.8, we obtain:

$$-\epsilon\mathbb{E}\langle\mathcal{K}_{f_\tau}D[f_\tau - f_*] + \frac{\lambda}{N}f_\tau, \mathcal{K}D[f_\tau - f_*] + \frac{M_\beta\lambda}{N}f_\tau\rangle_{\mathcal{H}}$$
$$\leq -\epsilon\frac{\lambda M_\beta + 1}{M_\beta N}EV(f_\tau, M_\beta\lambda) + 2\epsilon M_\beta(M_\beta^2 - 1)\frac{\lambda^2}{N^2}R^2.$$

Substituting it into the bound on $V(f_{\tau+1}, M_\beta\lambda)$ gives:

$$\mathbb{E}V(f_{\tau+1}, M_\beta\lambda)$$
$$\leq \big(1 - \epsilon(\frac{1+M_\beta\lambda}{M_\beta N} - 2\epsilon)\big)V(f_\tau, M_\beta\lambda) + \frac{2\epsilon\lambda}{N}\Big(M_\beta(M_\beta^2 - 1)\frac{\lambda}{N} + \epsilon\big(\frac{2\lambda}{N^2} + M_\beta(1 + \frac{2\lambda^2}{N^2})\big)\Big)R^2$$
$$\leq \big(1 - \epsilon\frac{1+M_\beta\lambda}{2M_\beta N}\big)V(f_\tau, M_\beta\lambda) + \frac{2\epsilon\lambda}{N}\Big(M_\beta(M_\beta^2 - 1)\frac{\lambda}{N} + \epsilon\big(\frac{2\lambda}{N^2} + M_\beta(1 + \frac{2\lambda^2}{N^2})\big)\Big)R^2,$$

which yields

$$\mathbb{E}V(f_T, M_\beta\lambda) \leq \frac{R^2}{2N}e^{-\frac{1+M_\beta\lambda}{2M_\beta N}T\epsilon} + \frac{4M_\beta\lambda}{1+M_\beta\lambda}\Big(M_\beta(M_\beta^2 - 1)\frac{\lambda}{N} + \epsilon\big(\frac{2\lambda}{N^2} + M_\beta(1 + \frac{2\lambda^2}{N^2})\big)\Big)R^2,$$

where we used the bound

$$V(f_0, M_\beta\lambda) = V(0, M_\beta\lambda) = \frac{1}{2N}\|f_*(\mathbf{x}_N)\|^2_{\mathbb{R}^N} - C_{M_\beta\lambda} \leq \frac{R^2}{2N}.$$

Next, we use the following inequality:

$$V(f) = L(f) - \min L(f) \leq L(f, M_\beta\lambda) - \min L(f, M_\beta\lambda) + \min L(f, M_\beta\lambda) - \min L(f)$$
$$\leq V(f, M_\beta\lambda) + L(f_*, M_\beta\lambda) - L(f_*) = V(f, M_\beta\lambda) + \frac{M_\beta\lambda}{2N}R^2,$$

which finally gives us the following bound on $\mathbb{E}V(f_T)$:

$$\mathbb{E}V(f_T) \leq \frac{R^2}{2N}e^{-\frac{1+M_\beta\lambda}{2M_\beta N}T\epsilon}$$
$$+ M_\beta\lambda\Big(\frac{1}{2N} + \frac{4M_\beta}{1+M_\beta\lambda}(M_\beta^2 - 1)\frac{\lambda}{N} + \frac{4\epsilon}{1+M_\beta\lambda}\big(\frac{2\lambda}{N^2} + M_\beta(1 + \frac{2\lambda^2}{N^2})\big)\Big)R^2.$$

$\square$

**Theorem G.23.** *(Theorem 3.8 in the main text) Let* $C = M_\beta\lambda\Big(\frac{1}{2N} + \frac{4M_\beta}{1+M_\beta\lambda}(M_\beta^2 - 1)\frac{\lambda}{N} + \frac{4\epsilon}{1+M_\beta\lambda}\big(\frac{2\lambda}{N^2} + M_\beta(1 + \frac{2\lambda^2}{N^2})\big)\Big)R^2$. *Assume that* $0 < \epsilon(\frac{\lambda}{N} + 1) < 1$ *and* $\frac{(1+M_\beta\lambda)}{4M_\beta N} \geq \epsilon$, $\frac{(1+\lambda)}{8N} \geq \epsilon$, $e^{\frac{4mC}{\beta}} \leq \frac{5}{4}$ *(this bound can be achieved by taking $\beta$ arbitrary large) and define* $T_1 = \Big[\frac{2M_\beta N}{\epsilon(1+M_\beta\lambda)}\log\frac{R^2}{2CN}\Big] + 1$. *Then* $\forall T \geq T_1$ *it holds that*

$$\mathbb{E}V(f_T, \lambda) \leq 2(C + \frac{\lambda}{N}R^2)e^{-\frac{1+\lambda}{4N}\epsilon(T-T_1)} + \frac{8\lambda}{1+\lambda}\Big(\frac{\lambda M_\beta^2}{N} + \epsilon\big(1 + \frac{2\lambda(1+\lambda)}{N^2}\big)\Big)R^2.$$

*Proof.* First, we apply the previous theorem to obtain a bound on $V(f_\tau)$ which we will use to claim that the kernels $\mathcal{K}_{f_\tau}$ and $\mathcal{K}$ are close to each other in expectation. If we take

$$T_1 = \Big[\frac{2M_\beta N}{\epsilon(1+M_\beta\lambda)}\log\frac{R^2}{2CN}\Big] + 1,$$

then the following inequalities hold $\forall\tau \geq T_1$:

$$EV(f_\tau) \leq 2C,$$

$$EV(f_\tau, \lambda) \le 2(C + \frac{\lambda}{N}R^2).$$

Then, analogously with the previous theorem, we estimate:

$$EV(f_{\tau+1}, \lambda) \le (1 + 2\epsilon^2)\mathbb{E}V(f_\tau, \lambda) - \epsilon\mathbb{E}\langle \mathcal{K}_{f_\tau}D[f_\tau - f_*]$$
$$+ \frac{\lambda}{N}f_\tau, \mathcal{K}D[f_\tau - f_*] + \frac{\lambda}{N}f_\tau\rangle + \frac{2\epsilon^2\lambda}{N}\Big(1 + \frac{2\lambda(1+\lambda)}{N^2}\Big)R^2.$$

Further, we bound $-\epsilon\mathbb{E}\langle \mathcal{K}_{f_\tau}D[f_\tau - f_*] + \frac{\lambda}{N}f_\tau, \mathcal{K}D[f_\tau - f_*] + \frac{\lambda}{N}f_\tau\rangle$ by Lemma G.19, instead of Lemma G.17, which we used in the previous theorem:

$\forall \tau \ge T_1$

$$-\mathbb{E}\langle \mathcal{K}_{f_\tau}D[f_\tau - f_*] + \frac{\lambda}{N}f_\tau, \mathcal{K}D[f_\tau - f_*] + \frac{\lambda}{N}f_\tau\rangle$$

$$\le \mathbb{E}\left((e^{\frac{2mV(f_\tau)}{\beta}} - 1) - \frac{1}{2}\right)\|\nabla_{\mathcal{H}}V(f_\tau, \lambda)\|_{\mathcal{H}}^2 + \mathbb{E}(e^{\frac{2mV(f_\tau)}{\beta}} - 1)^2\frac{\lambda^2}{2N^2}\|f_\tau\|_{\mathcal{H}}^2$$

$$\le 2\left(e^{\frac{2m\mathbb{E}V(f_\tau, \lambda)}{\beta}} - \frac{3}{2}\right)\frac{1+\lambda}{N}\mathbb{E}V(f_\tau, \lambda) + \frac{2\lambda^2 M_\beta^2}{N^2}R^2$$

$$\le \left(2\frac{1+\lambda}{N}e^{\frac{4mC}{\beta}} - 3\frac{1+\lambda}{N}\right)\mathbb{E}V(f_\tau, \lambda) + \frac{2\lambda^2 M_\beta^2}{N^2}R^2$$

$$\le \left(2\frac{1+\lambda}{N}e^{\frac{4mC}{\beta}} - 3\frac{1+\lambda}{N}\right)\mathbb{E}V(f_\tau, \lambda) + \frac{2\lambda^2 M_\beta^2}{N^2}R^2$$

$$\le -\frac{1+\lambda}{2N}\mathbb{E}V(f_\tau, \lambda) + \frac{2\lambda^2 M_\beta^2}{N^2}R^2.$$

Substituting this in the formula, we get:

$\forall \tau \ge T_1$

$$\mathbb{E}V(f_{\tau+1}, \lambda) \le \left(1 - \epsilon\big(\frac{1+\lambda}{2N} - 2\epsilon\big)\right)\mathbb{E}V(f_\tau, \lambda) + \frac{2\epsilon\lambda}{N}\Big(\frac{\lambda M_\beta^2}{N} + \epsilon\big(1 + \frac{2\lambda(1+\lambda)}{N^2}\big)\Big)R^2$$

$$\le (1 - \epsilon\frac{1+\lambda}{4N})\mathbb{E}V(f_\tau, \lambda) + \frac{2\epsilon\lambda}{N}\Big(\frac{\lambda M_\beta^2}{N} + \epsilon\big(1 + \frac{2\lambda(1+\lambda)}{N^2}\big)\Big)R^2.$$

Iterating the bound yields

$$\mathbb{E}V(f_T, \lambda) \le \mathbb{E}V(f_{T_1}, \lambda)e^{-\frac{1+\lambda}{4N}\epsilon(T-T_1)} + \frac{8\lambda}{1+\lambda}\Big(\frac{\lambda M_\beta^2}{N} + \epsilon\big(1 + \frac{2\lambda(1+\lambda)}{N^2}\big)\Big)R^2$$

$$\le 2(C + \frac{\lambda}{N}R^2)e^{-\frac{1+\lambda}{4N}\epsilon(T-T_1)} + \frac{8\lambda}{1+\lambda}\Big(\frac{\lambda M_\beta^2}{N} + \epsilon\big(1 + \frac{2\lambda(1+\lambda)}{N^2}\big)\Big)R^2.$$

$\square$

**Corollary G.24.** *(Convergence to the solution of the KRR / Convergence to the Gaussian Process posterior mean function). Under the assumptions of both previous theorems we have that:*[10]

$$\mathbb{E}\|f_T - f_*^\lambda\|_{L_2}^2 \le \max_{x \in \mathcal{X}}\mathcal{K}(x,x)\Big(4N(C + \frac{\lambda}{N}R^2)e^{-\frac{1+\lambda}{4N}\epsilon(T-T_1)}$$

$$+ \frac{16N\lambda}{1+\lambda}\Big(\frac{\lambda M_\beta^2}{N} + \epsilon\big(1 + \frac{2\lambda(1+\lambda)}{N^2}\big)\Big)R^2\Big).$$

*Proof.* By Lemma D.7, $V(f, \lambda) = \frac{1}{2N}\|f(\mathbf{x}_N) - f_*^\lambda(\mathbf{x}_N)\|_{\mathbb{R}^N}^2 + \frac{\lambda}{2N}\|f - f_*^\lambda\|_{\mathcal{H}}^2$. Then, by the previous theorem, we get a bound on $\frac{1}{2N}\mathbb{E}\|f_T(\mathbf{x}_N) - f_*^\lambda(\mathbf{x}_N)\|_{\mathbb{R}^N}^2$, and by Lemmas G.10 and G.8, we majorize our $L_2$ norm by $\mathbb{R}_N$ norm. Lemma then follows.

$\square$

---

[10]When $N \to \infty$, $\mathcal{K}$ converges to a certain kernel. Thus, $\max_{x \in \mathcal{X}}\mathcal{K}(x,x)$ can be estimated with a constant with probability arbitrary close to one.

**Lemma G.25** (Lemma 4.1 in the main text). *The following convergence holds almost surely in $x \in X$:*

$$h_T(\cdot) \xrightarrow[T \to \infty]{} \mathcal{GP}\left(\mathbb{0}_{L_2(\rho)}, \mathcal{K}\right).$$

*Proof.* From (5), we have that the covariance of $h_T$ is $\mathcal{K}$ independently from $T$. Thus, it remains to show that the limit is Gaussian almost surely which essentially holds due to the central limit theorem almost surely in $x \in X$:

$$h_T(x) = \frac{1}{\sqrt{T}} \sum_{i=1}^{T} h_{T,i}(x) \to \mathcal{N}\left(\mathbb{0}_{L_2}, \mathcal{K}(x,x)\right),$$

where each individual tree $h_{T,i}$ is centered i.i.d. (with the same distribution as $h_1$). $\qquad\square$

## H  IMPLEMENTATION DETAILS

In the experiments, we fix $\sigma = 10^{-2}$ (scale of the kernel) and $\delta = 10^{-4}$ (scale of noise), which theoretically can be taken arbitrarily. As a hyperparameter (that is estimated on the validation set), we consider $\beta \in \{10^{-2}, 10^{-1}, 1\}$. We use the standard CatBoost library and add the Gumbel noise term in selecting the trees for the "L2" scoring function, which is implemented in CatBoost out of the box but is not used by SGB and SGLB since it is not the default one for the library. Moreover, we do not consider subsampling of the data (as SGLB does also), and differently from SGB and SGLB, we disable the "boost-from-average" option. Finally, we set $l2-leaf-reg$ value to $0$, as SGLB does.

