# OpenReview forum: "Gradient Boosting Performs Gaussian Process Inference"
_ICLR.cc/2023/Conference — ICLR 2023 poster_

### Official Review · Reviewer_1ARD · 2022-10-21

**Confidence:** 3
**Correctness:** 4
**Technical Novelty And Significance:** 2
**Empirical Novelty And Significance:** 1
**Recommendation:** 6

**Clarity, Quality, Novelty And Reproducibility:**

        The paper is overall difficult to follow. It is written in an obfuscated way. For the reader non-familiar with Gaussian process nor Gradient boosting it is completely unreadable.


**Strength And Weaknesses:**

Strengths:

        - Nice theoretical results showing that gradient boosting can be understood as a kernel method.

Weaknesses:

        - The main result is a bit obvious since gradient boosting is minimizing objectives in functional space.

        - The fact that KGB performs better questions why not using a Gaussian process as the learning method.

        - The experimental section is weak. No error bars are given and therefore it is not possible to assess the significance of the results.



**Summary Of The Paper:**


        This paper analyzes gradient boosting ensembles and shows that they can be understood as a kernel method that is indeed finding the solution to an optimization problem that converges to the posterior mean of a Gaussian process. Using the technique known as sample-first-then-optimize the considered method can be used to generate samples from the posterior distribution of a Gaussian process. This means that output uncertainty can be readily obtained and used, for example for out of distribution detection. The proposed method is evaluated on several datasets from the UCI repository.


**Summary Of The Review:**

   I believe this is a nice paper showing a nice theoretical results. However, it is impaired by the questionable applications. In particular, I wonder why not using directly a Gaussian process if KGB is precisely approximating that. The experimental section is also weak and does not allow to check if the method proposed is significant. In particular, no error bars are given.

Minor:

Equations are not numbered and cannot be referred. There is a typo in the Eq. for the posterior variance of the GP.

N (f (x), σ 2 (x) is missing ).

---

> ### Author Response · Authors · 2022-11-09
> **Response to Reviewer 1ARD**
>
> Thank you very much for your feedback! We address your questions and concerns below.
>
> > The main result is a bit obvious since gradient boosting is minimizing objectives in functional space.
>
> We cannot agree with this statement. As we show, the kernel structure of the dynamics evolves with iterations. Therefore, the GP/KM interpretation is not obvious due to the dependency of the kernels on current approximation during inference. One of the main results of our work is to show that this dynamics stabilizes with iterations and leads to such interpretation due to the stabilization. It is easy to propose a gradient-boosting approach that is not interpretable in this way: consider a distribution of trees that depends on the current approximation and does not converge with iterations, e.g., in the $\beta \to 0$ limit.
>
> > The fact that KGB performs better questions why not using a Gaussian process as the learning method.
>
> Computing the kernel is not tractable as it requires summing up among all possible trees structures number of which grows as $(nd)^m$ which is unfeasible, not to mention the requirement to inverse the kernel matrix, which requires $\mathcal{O}(N^{2+\omega})$ number of operations. We write about this in Section 5, page 9. Our experiments are performed on large datasets, e.g., Year contains more than 500K samples. Also, our method can be used for multidimensional problems, thus making typical GP learning unfeasible.
>
> > The experimental section is weak. No error bars are given and therefore it is not possible to assess the significance of the results.
>
> Following Malinin et al. (2021), we perform cross-validation to estimate statistical significance with the paired t-test and highlight the approaches that are insignificantly different from the best one (p-value > 0.05). We discuss this in Section 5, page 9. Thus, all bold results are significant on level p < 0.05. Does this address your concern?
>
> > The paper is overall difficult to follow.
>
> We will do our best to improve the clarity of the text and will upload the updated paper. We will be very grateful if you specify any particular moments that need clarification.
>
> > Equations are not numbered and cannot be referred. There is a typo in the Eq. for the posterior variance of the GP. N (f (x), σ 2 (x) is missing ).
>
> We will number the equations in the revised version. Could you please specify which page this comment refers to?
>
> If there are any additional concerns, we will be happy to engage in further discussions while we are working on the revised version of the paper.

---

> > ### Comment · Reviewer_1ARD · 2022-11-17
> > **Response to reviwers**
> >
> > I would like to thank the reviewers for their response. I have considered it and have risen a bit my score in consequence. Thank you for the clarifications. There are still some typos, so I would encourage the authors to check their submission again. See N (f (x), σ 2 (x) in the first paragraph of section 2.2, and the definition of ~K(x,x) in lemma 2.1 which is having and extra ')' at the end.

---

> > > ### Author Response · Authors · 2022-11-19
> > > **Thanks**
> > >
> > > Thank you for the positive feedback and for pointing us to some typos. We corrected those and proofread the text.

---

### Official Review · Reviewer_HjFB · 2022-10-25

**Confidence:** 3
**Correctness:** 4
**Technical Novelty And Significance:** 3
**Empirical Novelty And Significance:** 3
**Recommendation:** 6

**Clarity, Quality, Novelty And Reproducibility:**

I would suggest the paper needs some editing to improve its readability, to communicate the main ideas more intuitively, without excessive mathematical baggage, and to clarify the particular theoretical contribution.

The relationship to other work could be expanded in the main text. The ‘related work’ section in the appendix does mention a few other kernel interpretations of tree-based methods, but it is fairly cursory.

The mathematical detail does not always differentiate between novel theory and restatements of standard results in GP learning / kernel methods.  It is unclear why symmetric trees are used. Are they essential ingredients of this procedure, or could other tree-building procedures work well, but just less readily admit mathematical analysis?  The definition of the RKHS inner product is not the standard one, is not cited, is highly technical, and its relevance is unclear. What are the key novel technical developments in the proof, for those of us not intimately familiar with this branch of the literature?  How can we understand the different terms in the convergence theorems?

There are a few places where unclear notation is used. For example, in section 2.3, the term $\min_{f\in H}$ appears without explanation in defining the loss function.

Finally, the title is a bit misleading. Standard gradient boosting does not perform Gaussian process inference, but can be adapted to do so.


**Strength And Weaknesses:**

Strengths
The paper presents a compelling connection between gradient boosting and Gaussian Process inference.  It presents a tree-based kernel, which is used to define a RKHS and the definition of the Gaussian process.  The paper presents experimental results that show impressive performance both for prediction and uncertainty quantification on a range of datasets. The paper is addressing an important and unsolved problem in a principled way.

Weaknesses
The paper would benefit from editing and condensing to make it clearer and to provide more intuition to the reader. The mathematical exposition is provided without much explanation or connection to empirical results, other than ensuring convergence. For example, the authors show some empirical improvement on bagged SGB and SGLB, but it’s not clear where the relative weakness of those methods stem from.

The experiments could be further strengthened in a way that both bolsters the case for the method, and improves the paper’s arguments.  For example, the authors do not compare to GP with standard kernels. This would be interesting to see, given the close connection to these methods. What is the dimensionality of these problems?  Does the method work well in high-dimensional problems, especially with sparsity?


**Summary Of The Paper:**

A method for posterior inference with a variant of GBDT is proposed. The weak learner used in this paper is a variant of the standard decision tree, where decision rules are oblivious (same split used at all nodes of the tree at the same level), the splitting procedure is randomized by use of the max-Gumbel trick to sample the decision rule. Shrinkage is applied throughout the training process as a regularizer.

The proposed inference technique is a form of sample-then-optimize, wherein a posterior sample is acquired by first sampling from a prior on the function space, then applying GBDT to the residuals of the sample from the prior and the training data.

The paper provides theoretical justification for their method, including finite-sample convergence of the gradient boosting algorithm, and convergence of the sampler to the posterior distribution.


**Summary Of The Review:**

The connection between boosting and GP inference is a compelling unification of different approaches to uncertainty quantification.  The paper is theoretically grounded and shows strong empirical results. The presentation could be made clearer, with more explanation of the significance of the theory, and intuition behind the performance of the method.

---

> ### Author Response · Authors · 2022-11-09
> **Response to Reviewer HjFB (part 2)**
>
> > The mathematical detail does not always differentiate between novel theory and restatements of standard results in GP learning / kernel methods.
>
> The theoretical novelty is the kernel structure of gradient boosting inference and convergence result under the assumption that kernels change during the iterations of the dynamics. All previous results regarding GP convergence typically assume that a kernel is stationary. Thus our work presents the convergence result under the particular case of dynamically changing kernels.
>
> > It is unclear why symmetric trees are used. Are they essential ingredients of this procedure, or could other tree-building procedures work well, but just less readily admit mathematical analysis?
>
> We use symmetric trees for clarity as arbitrary trees are harder to formalize in the algorithm’s description and to write down analytically the probabilities of trees for each iteration. However, our results apply to arbitrary trees. We mention the general case in footnote 3.
>
> > The definition of the RKHS inner product is not the standard one, is not cited, is highly technical, and its relevance is unclear. What are the key novel technical developments in the proof, for those of us not intimately familiar with this branch of the literature? How can we understand the different terms in the convergence theorems?
>
> We have changed the definition of the RKHS inner product to the standard one; we will upload a revised version soon. Our definition is equivalent to the standard one. Our key novel development in the proofs is considering the convergence of dynamically changing kernel structure. A key insight is that this dynamic effectively stabilizes around the stationary kernel. While dynamically changing kernels appear, e.g., in Deep Learning, our dynamic is different. In Deep Learning, kernels stabilize in the width limit, while in our case, kernels stabilize with iterations of inference. Such a case has not been considered in the literature (to the best of our knowledge) and poses technical challenges, e.g., it is not obvious that kernels stabilize and not obvious that images of all kernels are the same, which are cornerstones of our proofs. Different terms in convergence theorems that appear in bounds from left to right are: exponentially decaying term corresponding to gradient descent dynamic, error contribution from stochasticity, and error contribution from kernels dynamic. We will add this explanation in the revisited version.
>
> > There are a few places where unclear notation is used. For example, in section 2.3, the term $\mathrm{min}_{f \in \mathcal{H}}$ appears without explanation in defining the loss function.
>
> This term means that our problem is to optimize the loss function, i.e., $L \to \mathrm{min}$ means that we are seeking for minimizers of $L$ among $f \in \mathcal{H}$.
>
>
> > Finally, the title is a bit misleading. Standard gradient boosting does not perform Gaussian process inference, but can be adapted to do so.
>
> As we mention in the abstract, the standard gradient boosting (without modifications) converges to Gaussian Process’ posterior mean.
>
> If there are any additional questions or concerns, we will be happy to engage in further discussions while we are working on the revised version of the paper.

---

> ### Author Response · Authors · 2022-11-09
> **Response to Reviewer HjFB (part 1)**
>
> Thank you very much for the detailed feedback! Below we reply to your questions and concerns.
>
> > The paper would benefit from editing and condensing to make it clearer and to provide more intuition to the reader.
>
> We will do our best to improve the clarity of the text and will upload the updated paper. We will address the comments from the review (see below) and make clarifications throughout the text. However, if you notice any additional moments that need clarification, we will be grateful if you specify them.
>
> > the authors show some empirical improvement on bagged SGB and SGLB, but it’s not clear where the relative weakness of those methods stem from.
>
> Most of our results are concerned with the convergence of the method, which is tricky due to the dynamically changing kernels. We need the convergence result to claim the correspondence of GP and Gradient Boosting and derive our algorithm for posterior sampling. We will add this remark to our text to improve readability and clarity. We believe that the weaknesses of SGB come from the fact that the limiting distribution concentrates on the minimum of RMSE obtained via the Gradient Flow-like dynamics (which is studied in [1]), effectively making it unsuitable for knowledge uncertainty estimation. SGLB, on the other hand, samples from a similar distribution but with another kernel. We believe that our method outperforms SGLB due to its fast non-asymptotic convergence, while for SGLB we can only say that the convergence cannot be faster than those of the Euler-Maryama method, which is slow compared to the convergence rate of our method. We will expand the last sentences of Section 5 with these explanations.
>
> [1] Dombry, C., Duchamps, J. J. Infinitesimal gradient boosting. arXiv preprint arXiv:2104.13208. 2021.
>
> > The experiments could be further strengthened in a way that both bolsters the case for the method, and improves the paper’s arguments. For example, the authors do not compare to GP with standard kernels. This would be interesting to see, given the close connection to these methods. What is the dimensionality of these problems? Does the method work well in high-dimensional problems, especially with sparsity?
>
> Gradient boosting best performs with dense features. However, if the features can be represented as categorical, then CatBoost (on which our experiments rely) transforms them to dense via “target-based” encoding. Thus, the answer to this question depends on the nature of sparse features. In some cases, it will work well, while in others, it may not. This holds for all gradient-boosting algorithms.
>
> We do not compare our algorithm with typical GP methods since our method can be applied to large-scale high-dimensional problems as our rates are dimension-free and scale well with the dataset size compared to typical GP methods. Our datasets can be large (e.g., Year has more than 500K samples), which prevents us from comparing GBDT with other GP methods. Let us also note that our method aims to improve knowledge uncertainty within the Gradient Boosting framework, which is the state-of-the-art method for tabular datasets.
>
> > I would suggest the paper needs some editing to improve its readability, to communicate the main ideas more intuitively, without excessive mathematical baggage, and to clarify the particular theoretical contribution.
>
> We will improve the clarity of our theoretical contributions in the text. Our main theoretical contribution is to finalize the unification of all ML methods under the umbrella of Kernel Methods and Gaussian Processes. Similar results exist for Deep Learning with Neural Tangent Kernels. Furthermore, we provide non-asymptotic results on linear convergence of gradient boosting in a realistic setting. Surprisingly, it is a novel thing in the field.
>
> > The relationship to other work could be expanded in the main text. The ‘related work’ section in the appendix does mention a few other kernel interpretations of tree-based methods, but it is fairly cursory.
>
> There are indeed interesting works on kernel interpretations of tree-based methods, e.g., [2]. While these results consider neural tree ensembles, we will add them to the related work section. Thanks for the suggestion!
>
> [2] Kanoh R, Sugiyama M. A Neural Tangent Kernel Perspective of Infinite Tree Ensembles. ICLR 2022.

---

### Official Review · Reviewer_iZU9 · 2022-10-31

**Confidence:** 3
**Correctness:** 4
**Technical Novelty And Significance:** 3
**Empirical Novelty And Significance:** 3
**Recommendation:** 6

**Clarity, Quality, Novelty And Reproducibility:**

I found the text overall clear and well-written. Legend of Fig 1 should be changed (there’s something wrong with the colors).
As an emergency reviewer, I couldn’t check the related literature, and cannot evaluate precisely the novelty wrt the previous works referred to as SGB and SGLB.
I did not check the github repository.

**Strength And Weaknesses:**

Strengths: the paper strikes a is a nice balance between theory and practical relevance of the proposed method. In particular, there are nice non-asymptotic results in section 4.1.
Experiments show interesting results in posterior variance estimation and OOD detection.

Weaknesses: I find it hard (even for a Bayesian) to fully get the methods proposed in section 4, esp. regarding Bayesian terms as prior, posterior, on Monte Carlo sampling.
-Providing transitions between subsections would help. The discussion after corollary 4.2 is difficult to read, probably because it is essentially made of equations/complexities. What do you conclude from it?
- Usually, Bayesian have a handle on their priors, eg with hyper parameters. Is it the case here? I find this subsection difficult to get.
- SOme diagram to relate the different quantities from the prior/posterior/asymptotic posterior would be helpful.

**Summary Of The Paper:**

The paper is a nice theoretical one which shows an equivalence between two seemingly unrelated ML methods: gradient boosting decision trees on hand, and kernel methods/Gaussian processes on the other. On top of this equivalence, it is shown that model uncertainty can be obtained according to the GP variance.

**Summary Of The Review:**

In summary, I find that the paper is making well-grounded contributions which I think should be of interest for the ML community working on uncertainty quantification.

---

> ### Author Response · Authors · 2022-11-09
> **Response to Reviewer iZU9**
>
> Thank you very much for your positive feedback and comments that will help us to improve the paper. Below we reply to your questions and concerns.
>
> > I find it hard (even for a Bayesian) to fully get the methods proposed in section 4, esp. regarding Bayesian terms as prior, posterior, on Monte Carlo sampling. Providing transitions between subsections would help.
>
> We will do our best to improve the clarity of the text and will upload the updated paper.
>
> > The discussion after corollary 4.2 is difficult to read, probably because it is essentially made of equations/complexities. What do you conclude from it?
>
> The main conclusion is that our convergence rate up to a constant multiplier matches the results on the convergence of SGD for finite-sum of strongly convex objectives. This justifies the rapid exponential convergence of boosting that we see in practice. We will improve the readability of this part.
>
> > Usually, Bayesian have a handle on their priors, eg with hyper parameters. Is it the case here?
>
> We assume that all our hyperparameters are fixed a priori. There is a major difference with a typical Bayesian setup: we cannot evaluate the likelihood as it is intractable in our case (summation over all possible trees to compute kernel + inversion of the kernel on large data sets), this leads to an inability to assume non-trivial distribution over hyperparameters in the current setup. In future work, we are planning to explore how to incorporate them.
>
> > Some diagram to relate the different quantities from the prior/posterior/asymptotic posterior would be helpful.
>
> Thank you for the suggestion. We plan to add a diagram explaining all kernels that appear and how they are related to the dynamics of the gradient boosting inference to the revised version of the paper.
>
> If there are any additional questions or concerns, we will be happy to engage in further discussions while we work on the revised version of the paper.

---

> > ### Comment · Reviewer_iZU9 · 2022-11-14
> > **Thanks for the replies!**
> >
> > Thanks a lot for the responses. I'm overall satisfied with your rebuttal regarding my questions which are essentially clarification ones, anyway.

---

### Comment · Area_Chair_fuHr · 2022-11-14
**please discuss**

Hi reviewers. The authors have responded. Can I please ask you to engage with their response by posting further questions/comments here?

---

### Author Response · Authors · 2022-11-16
**Revised paper**

We would like to thank the reviewers for their valuable comments. We’ve uploaded the revised version in which we address the raised concerns.

We made the following updates:
- Added clarifications throughout the text and transitions between sections to improve readability. See, e.g., the first paragraph of Section 4.
- Simplified the statements of the main theorems (Section 3.5).
- Added a diagram illustrating all the kernels (Appendix A).
- Extended Related Work (Appendix B).
- Fixed the legend of Figure 1.

If there are any additional suggestions or questions, we will be happy to address them.

Sincerely, Authors

---

### Decision · Program_Chairs · 2023-01-20

**Decision:**

Accept: poster

**Justification For Why Not Higher Score:**

The reviewers were consistent in their scores.

**Justification For Why Not Lower Score:**

The reviewers were consistent in their scores.

**Metareview: Summary, Strengths And Weaknesses:**

This paper proposes a connection between a specific type of gradient boosting and kernel ridge regression, then uses the well-known connection between kernel ridge regression and Gaussian process regression to propose a Bayesian sampling approach. As noted by a reviewer the paper is "theoretically grounded and shows strong empirical results". The non-asymptotic results are particularly useful. As noted by multiple reviewers and acknowledged by the authors, there are a number of places where the clarity of the presentation could be improved, which I expect the authors will be able to accomplish.

**Note From Pc:**

if the above contains the word "oral" or "spotlight" please see: "oral" presentation means -> notable-top-5% and "spotlight" means -> notable-top-25%. As stated in our emails, we are disassociating presentation type from AC recommendations